# OTTER: Data Efficient Language-Supervised Zero-Shot Recognition with Optimal Transport Distillation

**Bichen Wu**[1][*], **Ruizhe Cheng**[2][*], **Peizhao Zhang**[1], **Peter Vajda**[1], **Joseph E. Gonzalez**[2]
[1]Meta Reality Labs, [2]UC Berkeley
`{wbc,stzpz,vajdap}@fb.com,{chengruizhe,jegonzal}@berkeley.edu`

## Abstract

Traditional computer vision models are trained to predict a fixed set of predefined categories. Recently, natural language has been shown to be a broader and richer source of supervision that provides finer descriptions to visual concepts than supervised "gold" labels. Previous works, such as CLIP, use InfoNCE loss to train a model to predict the pairing between images and text captions. CLIP, however, is data hungry and requires more than 400M image-text pairs for training. The inefficiency can be *partially* attributed to the fact that the image-text pairs are noisy. To address this, we propose OTTER (**O**ptimal **T**ranspor**T** distillation for **E**fficient zero-shot **R**ecognition), which uses online entropic optimal transport to find a soft image-text match as labels for contrastive learning. Based on pretrained image and text encoders, models trained with OTTER achieve strong performance with only 3M image text pairs. Compared with InfoNCE loss, label smoothing, and knowledge distillation, OTTER consistently outperforms these baselines in zero-shot evaluation on Google Open Images (19,958 classes) and multi-labeled ImageNet 10K (10032 classes) from Tencent ML-Images. Over 42 evaluations on 7 different dataset/architecture settings x 6 metrics, OTTER outperforms (32) or ties (2) all baselines in 34 of them. Our source code is open sourced at `https://github.com/facebookresearch/OTTER`.

## 1 Introduction

In real-world image recognition tasks, input images come from a broad range of distributions, spanning tens of thousands of object categories unknown during training. It is thus important for computer vision models to generalize to a large number of visual concepts that may or may not be present in the training data. This problem is called zero-shot learning (ZSL), which aims to transfer knowledge from some known classes with training data to a much larger number of unfamiliar classes.

Previous works on ZSL have explored using attributes (Romera-Paredes & Torr, 2015; Akata et al., 2015; 2013), class hierarchy (Wang et al., 2018; Kampffmeyer et al., 2019), and pretrained word embeddings (Frome et al., 2013; Norouzi et al., 2014) to transfer knowledge from pretrained image representations to recognize new classes. Recently, natural language has been used as a powerful source of supervision for visual representation learning. (Desai & Johnson, 2020; Sariyildiz et al., 2020; Zhang et al., 2020; Jia et al., 2020) demonstrate the effectiveness of pretraining on image-text data. Among them, CLIP (Radford et al., 2021) applies natural language supervision to zero-shot image recognition. It collects an enormous dataset with over 400M image caption pairs from the Internet, and trains an image encoder and a text encoder jointly with a contrastive loss to maximize the cosine similarity of paired image and text embeddings. CLIP demonstrates good zero-shot classification results on a wide range of downstream image classification datasets. However, a main constraint of CLIP is that it requires over 400M image-text pairs for training. Collecting and training on such a huge dataset is very expensive. The inefficiency can be partially attributed to the fact that the training labels from image-text pairs are noisy. As shown in Figure 1, in a typical image-text dataset, we observe that images and captions are loosely correlated. It is very common that one caption (image) can potentially match several other images (captions), and the ground-truth pairing is

---

[*]Equal contribution

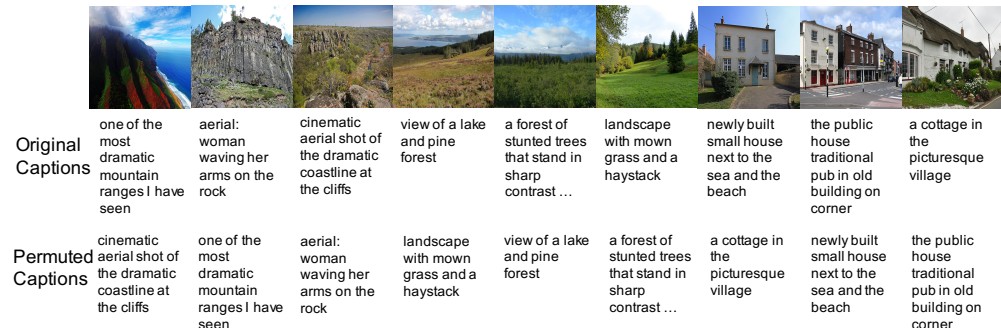

| | | | | | | | | |
|---|---|---|---|---|---|---|---|---|
| Original Captions | one of the most dramatic mountain ranges I have seen | aerial: woman waving her arms on the rock | cinematic aerial shot of the dramatic coastline at the cliffs | view of a lake and pine forest | a forest of stunted trees that stand in sharp contrast … | landscape with mown grass and a haystack | newly built small house next to the sea and the beach | the public house traditional pub in old building on corner | a cottage in the picturesque village |
| Permuted Captions | cinematic aerial shot of the dramatic coastline at the cliffs | one of the most dramatic mountain ranges I have seen | aerial: woman waving her arms on the rock | landscape with mown grass and a haystack | view of a lake and pine forest | a forest of stunted trees that stand in sharp contrast … | a cottage in the picturesque village | newly built small house next to the sea and the beach | the public house traditional pub in old building on corner |

Figure 1: Images and captions are only loosely correlated in many image-text datasets. The ground-truth pairing is not the only sensible match between texts and images. In the example above, we can find permutations of text captions that can still match with original images.

not the only sensible match. Note that examples in Figure 1 are not hand-picked special cases. In fact, such noisy image-text matching is prevalent in image-text datasets.

To quantitatively analyze this, we use a CLIP(Radford et al., 2021) VIT-B/32 pretrained on OpenAI's 400M dataset to estimate the matching probabilities between a batch of paired image-text samples. Specifically, we randomly sample 1000 batches from the CC3M (Sharma et al., 2018) and YFCC15M (subset of YFCC100M (Thomee et al., 2016)) datasets, and use the pretrained CLIP model to compute the image-to-text matching probabilities by taking the dot-product of the feature embeddings and taking a softmax along each row. For each batch, we compute three statistics (averaged across rows): default probability, non-default max probability, and non-default average probability. Note in both datasets, the matching probability between paired samples are far smaller than 1.0, and the probability decreases with the batch size. This indicates that there exist image and text samples that are not paired, but have nontrivial matching probabilities. This is further confirmed by the max matching probabilities between unpaired samples. In the extreme cases (CC 3M, 2048 batch size), the average of max matching probability between unpaired image-text samples is very close to the average of probability of paired samples. Despite prevalent noisy matching between images and texts, CLIP uses the InfoNCE loss (Hadsell et al., 2006) for training and uses the ground-truth pairings as hard labels. This ignores the many-to-many relationship within a batch of images and text captions, leading to noisy training signals and lower data efficiency.

Table 1: Matching probabilities estimated by CLIP on Conceptual Captions and YFCC

| Dataset | Batch Size | Paired | Unpaired Avg | Unpaired Max |
|---|---|---|---|---|
| CC 3M | 512 | 0.565 | 0.001 | 0.215 |
| | 1024 | 0.480 | 0.001 | 0.230 |
| | 2048 | 0.398 | 0.000 | 0.238 |
| YFCC 15M | 512 | 0.628 | 0.001 | 0.197 |
| | 1024 | 0.551 | 0.000 | 0.219 |
| | 2048 | 0.469 | 0.000 | 0.239 |

To address this, we propose OTTER, or **O**ptimal **T**ranspor**T** distillation for **E**fficient zero-shot **R**ecognition. We improve InfoNCE to consider the many-to-many relationship between unpaired images and texts. Specifically, given a batch of image and text tuples $\{(\mathbf{v}_i, \mathbf{t}_i)\}_{i=1:N}$, we first use image/text encoders to estimate a similarity matrix whose elements denotes similarity from image $\mathbf{v}_i$ to text caption $\mathbf{t}_j$. Based on the similarity matrix, we use optimal transport to find a matching probability between each possible image-text combination. To model the many-to-many relationship, we add an entropic regularization to the optimal transport so that the match is softly assigned. Entropic-regularized optimal transport can be solved efficiently with the iterative Sinkhorn-Knopp algorithm (Cuturi, 2013). Finally, we use the match as soft label to train the image and text encoders.

Based on pretrained image and text models, we use OTTER to train zero-shot models on the Conceptual Captions (CC) (Sharma et al., 2018), (subset of) Wikipedia-based Image Text (Srinivasan et al., 2021), and YFCC 15M (Thomee et al., 2016) datasets, which contain 3M, 5M, and 15M image-caption pairs, respectively. We evaluate the image encoder's zero-shot recognition of common visual concepts on Google Open Images (GOI) (Kuznetsova et al., 2020) (19,958 categories) and

multi-labeled ImageNet 10K (10032 categories) from Tencent-ML-Images (Wu et al., 2019a). Over 42 evaluations on 7 different dataset-architecture settings × 6 metrics, OTTER outperforms (32) or ties (2) all baselines in 34 of them. We also propose a quantitative vision-language compositionality benchmark and show comparable results to CLIP in Appendix D.

## 2 RELATED WORKS

**Zero-Shot Learning in Computer Vision**: Zero-shot learning (ZSL) studies the generalization of knowledge to unseen classes. Previous methods for zero-shot recognition in computer vision mainly follow three paradigms. The first type, including DeViSE (Frome et al., 2013) and ConSE (Norouzi et al., 2014), uses pretrained word embedding vectors to represent different categories and implicitly model their relationships. However, word embedding is a preliminary and limited representation of class relationships, which hurts performance. The second paradigm, including GCNZ (Wang et al., 2018), DPGZ (Kampffmeyer et al., 2019), and HZSL (Liu et al., 2020), explicitly models class relationships as a graph, and uses a graph convolutional network (GCN), or a predefined class hierarchy, such as WordNet (Feinerer & Hornik, 2020), to learn the knowledge propagation between classes. However, real-world class relationships are complicated and simple graph structures such as WordNet are too limited to model such relationships. Lastly, (Romera-Paredes & Torr, 2015; Akata et al., 2015; 2013) rely on human-labeled attributes to model semantics of classes. The scalability of these methods are limited by the need for attribute annotations. More recently, CLIP (Radford et al., 2021) applies language-supervision to ZSL by training on image caption pairs. Our work is based on CLIP and we generalize the InfoNCE loss to improve its data efficiency.

**Vision and Language**: Natural language has long been used as a source of supervision in fields like image-text retrieval Hironobu et al. (1999), object classification Wang et al. (2009), and video understanding (Ramanathan et al., 2013). Socher et al. (2014); Karpathy et al. (2014); Li et al. (2019); Chen et al. (2021a) have proposed methods of learning visual and language representations in a joint embedding space. More recently, (Lu et al., 2019; Chen et al., 2020c; Qi et al., 2020) propose using a cross-modal attention mechanism to increase performance in image-text matching. In addition, (Joulin* et al., 2016; Li et al., 2017; Desai & Johnson, 2020; Sariyildiz et al., 2020) demonstrate that good visual representations can be learned by predicting image captions. To scale up vision-language joint training, CLIP (Radford et al., 2021) and ALIGN (Jia et al., 2020) both collect their own image-text datasets with 400M and 1B image-caption pairs.

**Optimal transport** (OT) is a theory that enables comparison of two probability distributions whose supports may not overlap. OT has been applied to many areas such as domain adaptation (Courty et al., 2016), generative models (Salimans et al., 2018), and self-supervised vision representation learning (Caron et al., 2020; Asano et al., 2019). In vision and language, (Chen et al., 2020a) uses OT to align objects in images and words in texts. The problem formulation of our work is similar to (Damodaran et al., 2018), where OT is used to mitigate the label noise in remote-sensing data under supervised learning. In our paper, we extend the method from supervised learning to contrastive learning, where OT is a natural way to estimate pairings between images and texts. In another related work, (Chen et al., 2021b) adds an additional OT-based Wasserstein loss to contrastive representation distillation (Tian et al., 2019). The loss matches student representations to teacher representations in a batch. (Chen et al., 2021b) is different from our method since it directly minimizes the Wassertein loss between two models' representations, while our method uses OT to estimate the pairing probability and use the probability for knowledge distillation. Directly minimizing Wasserstein loss between image/text embeddings in our case will lead to collapsed representations, where models generate constant output regardless of inputs.

**Other related works:** Our work is also related to areas including learning with noisy labels, contrastive learning, and knowledge distillation. Our method uses OT to estimate the matching probability of unpaired images and texts. This is reminiscent to estimating label transition probability under noisy labels (Song et al., 2020). Our method is based on contrastive learning, which is commonly used in self-supervised visual representation learning (Chen et al., 2020b; He et al., 2020). For vision representation learning, (Robinson et al., 2020) argues that sampling hard negative pairs can improve learning efficiency. For language-supervised representation learning, however, it is important to mitigate the noise of widely spread hard negative samples, since positive image-text pairs are usually only loosely correlated. Our method is also an extension to knowledge distillation (KD) (Hinton et al., 2015). Typical KD directly relies on a teacher model to directly generate a target distribution (Xie et al., 2020; Bagherinezhad et al., 2018; Caron et al., 2021). Our method is

different since our target distribution is computed by OT based on the pairwise similarity estimated by a teacher model. Experiments show that this works better for image-text contrastive learning.

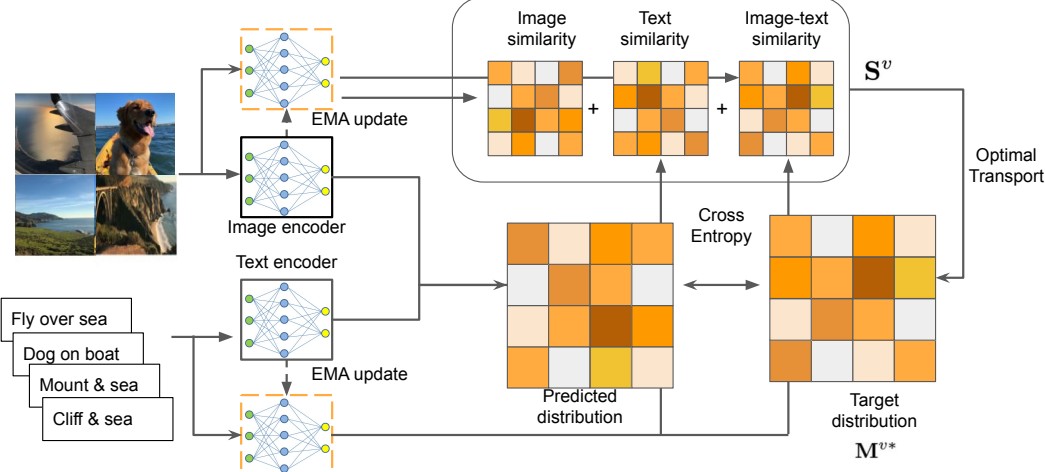

Figure 2: Architecture of OTTER. We use image and text embeddings to compute similarity matrices $\mathbf{S}^v$ (and $\mathbf{S}^t$), which is then used to solve for matching probabilities $\mathbf{M}^{v*}$ (and $\mathbf{M}^{t*}$) as targets.

## 3 METHODS

We introduce OTTER in this section. Let $\{(\mathbf{v}_i, \mathbf{t}_i)\}_{i=1:N}$ be a batch of paired image-text tuples sampled from data distribution $p(\mathbf{v}, \mathbf{t})$. Our model contains an image encoder $f_v(\cdot)$ and a text encoder $f_t(\cdot)$ that map image $\mathbf{v}_i$ and text $\mathbf{t}_i$ to $\ell_2$-normalized embeddings $\mathbf{z}_i^v$ and $\mathbf{z}_i^t$ respectively.

### 3.1 CONTRASTIVE LEARNING WITH INFONCE LOSS

CLIP (Radford et al., 2021) trains the image and text encoders with contrastive learning to pull the paired image and text embeddings closer, and push the unpaired embeddings farther. This is achieved by miminizing the InfoNCE loss $\mathcal{L}_{\text{InfoNCE}} = \mathcal{L}_v + \mathcal{L}_t$. $\mathcal{L}_v$ is the loss for matching images to text captions, and $\mathcal{L}_t$ is for text-to-image matching. $\mathcal{L}_v$ is defined as

$$\mathcal{L}_v = -\frac{1}{N}\sum_{i=1}^{N}\sum_{j=1}^{N} I_{ij} \log p_v(\mathbf{z}_i^v, \mathbf{z}_j^t; \tau) = -\frac{1}{N}\sum_{i=1}^{N}\sum_{j=1}^{N} I_{ij} \log \frac{\exp((\mathbf{z}_i^{v\top}\mathbf{z}_j^t)/\tau)}{\sum_{k=1}^{N}\exp((\mathbf{z}_i^{v\top}\mathbf{z}_k^t)/\tau)}, \quad (1)$$

where $(\mathbf{z}_i^{v\top}\mathbf{z}_j^t)$ is the cosine similarity between two $\ell_2$-normalized embedding vectors. $\tau$ is a (trainable) temperature parameter. $I_{ij}$ is the element of an identity matrix $\mathbf{I}_N$ with $I_{ii} = 1, \forall i$ and $I_{ij} = 0, \forall i \neq j$. Note $p_v$ is normalized across $\mathbf{z}_k^t$ for $k = 1, \cdots, N$ in the denominator. Symmetrically, we define $\mathcal{L}_t$ and $p_t$ in the same way as Equation (1) except we normalize across $\mathbf{z}_k^v$.

Equation (1) is a rather redundant way of writing the InfoNCE loss, as $I_{ij}$ are all zeros for unpaired image-text samples. However, this shows that InfoNCE is essentially the cross entropy between a one-hot distribution $I_{ij}$ and the estimated probability $p_v(\mathbf{z}_i^v, \mathbf{z}_j^t; \tau)$. One-hot distribution assumes that within a batch of images and text captions, the only match for image $\mathbf{v}_i$ is its paired text caption $\mathbf{t}_i$. However, as shown in Figure 1 and Table 1, this assumption is not true. Paired images and text captions are only loosely correlated. It is common that one image can match with several other texts and vice versa. The ground-truth match provided by the dataset is not the only sensible match between images and texts. One-hot labels are therefore noisy, leading to degraded learning performance.

### 3.2 MODELING THE PROBABILITY OF UNPAIRED IMAGE-TEXT MATCHING

To better capture the many-to-many relationship in image-text datasets, we modify InfoNCE in Equation (1) to consider the matching probability of unpaired images and texts. For a batch of $N$ image-text pairs, we define $Y_i \in \{1, \ldots, N\}$ as a random variable, and let $q_v(Y_i = j|\mathbf{v}_{1:N}, \mathbf{t}_{1:N})$ be the probability that image $\mathbf{v}_i$ should be matched with text caption $\mathbf{t}_j$ in the batch. We model this as

$$q_v(Y_i = j|\mathbf{v}_{1:N}, \mathbf{t}_{1:N}) = q_v(Y_i = i)I_{ij} + q_v(Y_i \neq i)M_{ij} = \alpha I_{ij} + (1-\alpha)M_{ij} \quad (2)$$

where $M_{ij} := q_v(Y_i = j | Y_i \neq i, \mathbf{v}_{1:N}, \mathbf{t}_{1:N})$, $M_{ii} = 0 \ \forall i$ is the conditional probability of image $\mathbf{v}_i$ being matched to $\mathbf{t}_j$ given that it is not matched to text $\mathbf{t}_i$. For simplicity, we write $q_i^v(j) := q_v(Y_i = j | \mathbf{v}_{1:N}, \mathbf{t}_{1:N})$. $\alpha \in [0, 1]$ is the prior probability that image $\mathbf{v}_i$ is matched with its paired text caption $\mathbf{t}_i$. $\alpha$ reflects the noise level in the dataset. In an ideal noiseless dataset, $\alpha = 1$, so $q_i^v(j) = I_{ij}$. This is the case where we should use the one-hot labels $I_{ij}$ for contrastive learning. However, in image-text datasets, it is common for an unpaired text caption $\mathbf{t}_j$ to be a better match for image $\mathbf{v}_i$, as shown in Table 1. In this case, $\alpha < 1$ and using $I_{ij}$ as the target distribution is no longer accurate. So we generalize the InfoNCE loss in Equation (1) by replacing $I_{ij}$ with the more generic $q_i^v(j)$ as

$$\mathcal{L}_v = -\frac{1}{N} \sum_{i=1}^{N} \sum_{j=1}^{N} [\alpha I_{ij} + (1 - \alpha) M_{ij}^v] \log p_v(\mathbf{z}_i^v, \mathbf{z}_j^t; \tau), \tag{3}$$

$\alpha I_{ij}$ provides supervision on paired image-text samples and $(1 - \alpha) M_{ij}^v$ supervises unpaired samples. The question is how do we estimate $M_{ij}^v$. A simple estimation is to let $M_{ij}^v = (1 - I_{ij})/(N - 1) \ \forall i, j$ be a uniform distribution. This is equivalent to the *label smoothing* method proposed in (Szegedy et al., 2016). However, this completely ignores the contents of images $\mathbf{v}_{1:N}$ and texts $\mathbf{t}_{1:N}$.

### 3.3 MODELING WITH OPTIMAL TRANSPORT

To design a better method of estimating $M_{ij}^v$, we start from two intuitions: first, in a reasonable image-text dataset, there are no bad images or texts. We assume all the images and texts are equally matchable so they should have equal matching probabilities. Second, the matching probability from image $\mathbf{v}_i$ to caption $\mathbf{t}_j$ should depend on their similarity estimation $S_{ij}$. A relatively higher similarity $S_{ij}$ should lead to higher matching probability $M_{ij}^v$. An estimation for $M_{ij}^v$ that satisfies the two intuitions can be obtained by solving the following entropic optimal transport problem (Cuturi, 2013)

$$\mathbf{M}^{v*} = \arg\max_{\mathbf{M} \in \mathcal{M}} \langle \mathbf{M}, \mathbf{S}^v \rangle_F + \lambda H(\mathbf{M}). \tag{4}$$

$\mathbf{S}^v \in \mathbf{R}^{N \times N}$ is a similarity matrix whose elements $S_{ij}^v$ denotes the similarity from $\mathbf{v}_i$ to $\mathbf{t}_j$. We discuss how to compute $\mathbf{S}^v$ in Section 3.4. $\langle \mathbf{M}, \mathbf{S}^v \rangle_F = \sum_{ij} M_{ij} S_{ij}^v$ is the Frobenius inner product between the similarity matrix $\mathbf{S}^v$ and the matching plan $\mathbf{M}$. Maximizing this term ensures $\mathbf{M}$ is similar to $\mathbf{S}^v$, *i.e.*, larger $S_{ij}^v$ leads to larger $M_{ij}$ and vice versa. Meanwhile, we add an entropy regularization on $\mathbf{M}$ as $H(\mathbf{M}) = -\sum_{ij} M_{ij} \log M_{ij}$. This ensures that $\mathbf{M}$ does not over concentrate on a few elements. We constrain the solution of Equation (4) to be a transportation polytope

$$\mathcal{M} = \{\mathbf{M} \in \mathbb{R}_+^{N \times N} \mid \mathbf{M} \mathbf{1}_N = \frac{1}{N} \mathbf{1}_N, \mathbf{M}^\top \mathbf{1}_N = \frac{1}{N} \mathbf{1}_N\}. \tag{5}$$

This constraint ensures that the solution $\mathbf{M}^{v*}$ satisfies the first intuition – all images and texts are equally important and should be matched with equal probabilities. Moreover, as proven in (Cuturi, 2013), the solution to Equation (4) takes the form of a normalized exponential matrix

$$\mathbf{M}^{v*} = \text{Diag}(\mathbf{r}) \exp(\mathbf{S}^v/\lambda) \text{Diag}(\mathbf{c}), \tag{6}$$

where $\mathbf{r}, \mathbf{c} \in \mathbb{R}^N$ are row and column normalization vectors and can be calculated through the iterative Sinkhorn-Knopp algorithm (Cuturi, 2013). The Sinkhorn-Knopp algorithm can be efficiently implemented on GPU and we provide a pseudo-code implementation in Appendix B.

From Equation (6), it is clear that $\mathbf{M}^{v*}$ satisfies our second intuition that a similarity $S_{ij}$ leads to higher matching probability since $M_{ij}^{v*} \sim \exp(S_{ij}^v/\lambda)$. The role of the entropic regularization is also clear. A larger $\lambda$ or higher entropy regularization and leads to "softer" distribution for $M_{ij}^{v*}$. On the other hand, a smaller $\lambda$ or lower entropy regularization leads to "harder" distribution for $M_{ij}^{v*}$.

### 3.4 COMPUTING THE SIMILARITY MATRIX

To compute the similarity from image $\mathbf{v}_i$ to text $\mathbf{t}_j$, we can use a pair of teacher encoders $\tilde{f}_v(\cdot), \tilde{f}_t(\cdot)$ to compute $\ell_2$-normalized embeddings $\tilde{\mathbf{z}}_i^v, \tilde{\mathbf{z}}_j^t$. Denoting $\tilde{\mathbf{Z}}^v, \tilde{\mathbf{Z}}^t \in \mathbf{R}^{d \times N}$ as matrcies whose columns are $\tilde{\mathbf{z}}_{1:N}^v, \tilde{\mathbf{z}}_{1:N}^t$ respectively, we compute the similarity matrix as

$$\mathbf{S}^v = \gamma_v \tilde{\mathbf{Z}}^{v\top} \tilde{\mathbf{Z}}^v + \gamma_t \tilde{\mathbf{Z}}^{t\top} \tilde{\mathbf{Z}}^t + \tilde{\mathbf{Z}}^{v\top} \tilde{\mathbf{Z}}^t - \eta \mathbf{I}_N. \tag{7}$$

The first term $\tilde{\mathbf{Z}}^{v\top} \tilde{\mathbf{Z}}^v \in \mathbf{R}^{N \times N}$ compares the image similarities, as $(\tilde{\mathbf{Z}}^{v\top} \tilde{\mathbf{Z}}^v)_{ij} = \tilde{\mathbf{z}}_i^{v\top} \tilde{\mathbf{z}}_j^v$ is the cosine similarity between image embeddings. Intuitively, it assumes that for a pair of similar images, it is likely that we can exchange their text captions. Similarly, $\tilde{\mathbf{Z}}^{t\top} \tilde{\mathbf{Z}}^t$ compares the text similarities.

It assumes that if a pair of text captions are similar, it is more likely that one text caption can match the other image. The term $\tilde{\mathbf{Z}}^{v\top}\tilde{\mathbf{Z}}^t$ considers the similarity between the image and text embeddings. Finally, $\eta\mathbf{I}_N$ with $\eta \to \infty$ ensures the diagonal terms of $\mathbf{S}^v$ are infinitely small. This effectively sets the diagonal terms of $\mathbf{M}^{v*}$ to 0, which is necessary since $M_{ij}$ is conditioned on $Y_i \neq i$.

There are several options to instantiate $\tilde{f}_v(\cdot)$ and $\tilde{f}_t(\cdot)$. The simplest option is to use the original image and text encoder $f_v(\cdot), f_t(\cdot)$ as $\tilde{f}_v(\cdot), \tilde{f}_t(\cdot)$. Alternatively, following recent works (He et al., 2020; Caron et al., 2021; Liu et al., 2021), $\tilde{f}_v(\cdot), \tilde{f}_t(\cdot)$ can share the same model architecture with $f_v(\cdot), f_t(\cdot)$, but their weights are updated as an exponential moving average as $\tilde{\theta} \leftarrow m\tilde{\theta} + (1-m)\theta$, where $\tilde{\theta}$ is the weight for $\tilde{f}_v(\cdot), \tilde{f}_t(\cdot)$, $\theta$ is the weight for $f_v(\cdot), f_t(\cdot)$, and $m$ is a momentum parameter set to 0.999. Of course, we can also use trained image and text encoders such as CLIP for $\tilde{f}_v(\cdot)$ and $\tilde{f}_t(\cdot)$. We adopt the first two options in our paper, since we want to avoid using extra image-text pairs.

### 3.5 RELATIONSHIP WITH KNOWLEDGE DISTILLATION

OTTER is an extension of conventional knowledge distillation (KD) (Hinton et al., 2015). Equation (3) computes the cross entropy $H(q_i^v, p_i^v)$ between $q_i^v(j)$ and $p_i^v(j) := p_v(\mathbf{z}_i^v, \mathbf{z}_j^t; \tau)$, where $q_i^v(j)$ is the teacher distribution solved by OT and $p_i^v(j)$ is the student distribution with logits $(\mathbf{z_i}^{v\top}\mathbf{z_j}^t)/\tau$ computed by $f(\cdot)_v, f(\cdot)_t$. A more conventional way to compute KD's teacher distribution is

$$q_v(\tilde{\mathbf{z}}_i^v, \tilde{\mathbf{z}}_j^t; \tau) = \frac{\exp((\tilde{\mathbf{z}}_i^{v\top}\tilde{\mathbf{z}}_j^t)/\tau)}{\sum_{k=1}^{N}\exp((\tilde{\mathbf{z}}_i^{v\top}\tilde{\mathbf{z}}_k^t)/\tau)}, \tag{8}$$

where $\tilde{\mathbf{z}}_i^v, \tilde{\mathbf{z}}_j^t$ are computed by the teacher $\tilde{f}_v(\cdot), \tilde{f}_t(\cdot)$. We can re-write Equation (8) in the matrix form as $\mathbf{Q}^v = \mathrm{Diag}(\mathbf{r})\exp(\tilde{\mathbf{Z}}^{v\top}\tilde{\mathbf{Z}}^t/\tau)\mathrm{Diag}(\mathbf{c})$, where $r_i = 1$, and $c_i = 1/\sum_{k=1}^{N}\exp((\tilde{\mathbf{z}}_i^{v\top}\tilde{\mathbf{z}}_k^t)/\tau)$. Note this teacher distribution has the same form as OTTER in Equation (6), but with two differences. First, OTTER's similarity matrix $\mathbf{S}^v$ in Equation (7) have three more terms: $\gamma_v\tilde{\mathbf{Z}}^{v\top}\tilde{\mathbf{Z}}^v, \gamma_t\tilde{\mathbf{Z}}^{t\top}\tilde{\mathbf{Z}}^t, \eta\mathbf{I}_N$. In comparison, KD ignores image-image, text-text similarities and does not exclude diagonal terms. By setting $\gamma_v = \gamma_t = \eta = 0$, their similarity matrices are equivalent. Second, OTTER's normalization vectors $\mathbf{r}, \mathbf{c}$ in Equation (6) are solved with Sinkhorn-Knopp while for KD $\mathbf{r}, \mathbf{c}$ are computed by a Softmax function. In fact, if we set the #iteration to 0 in Algorithm 2 (Appendix B), Sinkhorn-Knopp is equivalent to Softmax, as also noted by (Caron et al., 2021).

## 4 EXPERIMENTS

In this section, we discuss our experiments validating the effectiveness of OTTER. We open-sourced our code at `https://github.com/facebookresearch/OTTER`. To setup a baseline, we follow CLIP (Radford et al., 2021) to train an image and a text encoder to predict the pairing of image and text samples using the infoNCE loss. Since the dataset used by CLIP is not released, we train on three publicly available datasets, Conceptual Captions 3M (CC) (Sharma et al., 2018), Wikipedia-base Image-Text Dataset (WIT), and YFCC 15M (Thomee et al., 2016). We only train on images with English captions in all 10 partitions of the WIT dataset, resulting in 5M image-text pairs in total. Since the datasets we use are small ($\sim$100x smaller than the one used by CLIP), we have to use pre-trained models to initialize the image and text encoders. Also, due to the datasets' limited scale and concept coverage, models trained on CC or WIT do not perform well on domain-specific datasets such as Stanford Cars (Krause et al., 2013) and FGVC Aircraft (Maji et al., 2013). To test zero-shot recognition on *common visual concepts*, we evaluate our models on the test set of Google Open Image (GOI) (Kuznetsova et al., 2020), which contains 19,958 classes. We also evaluate on the test set of multi-labeled ImageNet 10K (10032 classes) dataset whose labels come from Tencent ML-Images (Wu et al., 2019a). Each image in ImageNet 10K is auto-labeled with highly-correlated class labels from GOI, alleviating the single-label issue of ImageNet 21K and 1K. To compare with previous ZSL methods (Norouzi et al., 2014; Frome et al., 2013; Wang et al., 2018; Liu et al., 2020), we report the ZSL performance of one of our models on ImageNet21K+1K.

**Training:** We adopt a training recipe similar to BiT's finetuning strategy (Kolesnikov et al., 2020): We use SGD with an initial learning rate of 3e-3, a cosine annealing scheduler, momentum 0.9, and no weight decay. Input images are resized to 256x256 and randomly cropped to 224x224 while test images are resized to 256x256 and center-cropped to 224x224. We train on 8 V100 GPUs using Pytorch (Paszke et al., 2019) distributed data parallel with a total batch size of 512 (64 per GPU) for

10 epochs. While CLIP (Radford et al., 2021) computes InfoNCE using sub-batches on each GPU, we gather logits from all GPUs for OTTER and baselines.

**Inference:** For inference, we follow CLIP to compute the text embeddings for the target classes using the trained text encoder, and we use a prompt template of "a photo of {label}" to augment the label texts. Next, we fit a KNN using the text embeddings. Given an image, we find the top K nearest label embedding neighbors to the image embedding based on cosine similarity.

**Evaluation:** GOI (Kuznetsova et al., 2020) and ImageNet 10K from Tencent-ML-Images (Wu et al., 2019a) are multi-labeled. Following previous work on ZSL (Norouzi et al., 2014; Frome et al., 2013; Wang et al., 2018; Liu et al., 2020), we use flat hit @ k (FH@K) for evaluation. FH@K is the percentage of test images such that the top K predictions of the model overlap with true labels and is formally defined as $\frac{1}{N}\sum_{i=1}^{N}\mathbb{1}(\{\{f(\mathbf{v}_i)\}_K \cap L_i \neq \varnothing\})$, where $\{f(\mathbf{v}_i)\}_K$ is the top K predictions for the $i$-th image and $L_i$ is the set of true labels.

Table 2: FH@K on test sets of Google Open Images and ImageNet10K from Tencent-ML-Images.

| Data | Image encoder | Text encoder | Method | GOI FH@K (%) | | | IN10K FH@K (%) | | |
|---|---|---|---|---|---|---|---|---|---|
| | | | | 1 | 5 | 10 | 1 | 5 | 10 |
| CLIP (400M) | ResNet50 ViT-B/32 | CLIP Transformer | InfoNCE | 26.5 27.5 | 54.0 55.3 | 64.3 65.4 | 20.1 22.5 | 44.8 49.1 | 56.4 60.7 |
| CC (3M) | Wide ResNet50x2 | | InfoNCE | 28.6 | 58.6 | 69.8 | 11.0 | 29.9 | 40.6 |
| | ResNet50 | | InfoNCE | 26.8 | 55.1 | 66.4 | 10.9 | 29.4 | 40.5 |
| | | | LS | 26.3 | 55.9 | 67.5 | 10.1 | 29.6 | 39.8 |
| | | | KD | 26.7 | 55.3 | 67.1 | 10.0 | 27.5 | 38.5 |
| | | | OTTER | **29.1** | **59.6** | **70.9** | **12.0** | **31.8** | **42.1** |
| | ResNet34 | DeCLUTR -Sci-base | InfoNCE | 22.8 | 50.0 | 61.5 | 7.9 | 23.7 | 33.0 |
| | | | LS | 19.8 | 46.9 | 59.2 | 6.7 | 21.9 | 31.9 |
| | | | KD | 21.1 | 47.9 | 59.8 | 7.3 | 23.0 | 32.5 |
| | | | OTTER | **24.2** | **52.6** | **64.4** | **9.0** | **25.6** | **35.4** |
| | FBNetV3-A | | InfoNCE | 27.2 | 57.0 | 69.0 | 10.0 | 27.9 | 38.5 |
| | | | LS | 24.2 | 53.9 | 65.7 | 8.9 | 26.7 | 38.0 |
| | | | KD | 26.9 | 56.7 | 68.4 | **10.7** | 28.9 | 39.7 |
| | | | OTTER | **27.5** | **57.2** | 69.0 | 10.4 | **29.4** | **39.9** |
| | FBNetV3-C | | InfoNCE | 25.7 | 54.3 | 66.1 | 8.7 | 25.8 | 35.8 |
| | | | LS | 24.8 | 54.0 | 66.1 | 9.7 | 26.8 | 37.6 |
| | | | KD | 26.6 | 55.8 | 67.6 | **10.5** | 28.2 | 38.9 |
| | | | OTTER | **27.5** | **57.6** | **69.1** | 10.4 | **28.7** | **39.4** |
| | ResNet50 | Sentence -BERT-base | InfoNCE | 25.5 | 52.2 | 62.8 | 9.5 | 26.1 | 35.9 |
| | | | LS | 24.5 | 50.8 | 61.6 | 9.3 | **26.7** | **37.0** |
| | | | KD | 25.6 | 52.3 | 62.4 | 9.8 | 26.2 | 36.0 |
| | | | OTTER | **26.1** | **53.1** | **63.4** | **9.9** | 26.6 | 36.6 |
| WIT (5M) | ResNet50 | DeCLUTR -Sci-base | InfoNCE | 13.5 | 34.0 | 44.8 | 6.3 | 19.2 | 27.8 |
| | | | LS | 14.3 | 35.5 | 46.2 | **6.4** | 19.8 | 28.9 |
| | | | KD | 14.4 | 35.0 | 45.9 | 6.2 | 19.3 | 28.0 |
| | | | OTTER | **14.5** | **36.4** | **47.7** | 6.2 | 19.8 | **29.0** |
| YFCC (15M) | ResNet50 | DeCLUTR -Sci-base | InfoNCE | 18.8 | 42.9 | 53.6 | 8.9 | 26.3 | 36.9 |
| | | | LS | 19.6 | 44.9 | 55.7 | **9.8** | **28.2** | **38.8** |
| | | | KD | 19.5 | 43.5 | 54.2 | 8.9 | 26.0 | 36.7 |
| | | | OTTER | **20.6** | **45.4** | **55.9** | 9.3 | 27.4 | 38.1 |

## 4.1 COMPARING OTTER WITH BASELINES

To compare with OTTER, we include three baselines: 1) InfoNCE with hard labels; 2) InfoNCE with label-smoothing (LS) (Szegedy et al., 2016), as described in Section 3.2; 3) InfoNCE with knowledge distillation (KD) (Hinton et al., 2015), as described in Section 3.5. In addition to the experimental setting described above, we use the following OTTER hyper-parameters: we set the loss coefficient $\alpha = 0.5$, set $\gamma_v = \gamma_t = 1$ for the similarity matrix. We use the exponential-moving average (EMA) of the image/text encoders as teachers and set the EMA decay to 0.999. For Sinkhorn-Knopp, we set

Table 3: Flat hit @K on ImageNet 21K+1K.

| Dataset | Image Encoder | Text Encoder | Method | Flat Hit@k(%) | | | |
|---------|---------------|--------------|--------|------|------|------|------|
| | | | | 1 | 2 | 5 | 10 |
| IN1k (1.2M) | ResNet50 | skip-gram | DeViSE | 0.3 | 0.9 | 2.2 | 3.6 |
| | | skip-gram | ConSE | 0.1 | 1.5 | 3.5 | 4.9 |
| | | GloVe | GCNZ | 1.0 | 2.3 | 5.3 | 8.1 |
| | | GloVe | HZSL | 2.2 | 4.6 | 9.2 | 12.7 |
| CC (3M) | FBNetV3-C | DeCLUTR-Sci-base | InfoNCE | 3.2 | 4.8 | 8.8 | 12.9 |
| | | | LS | 3.4 | 5.1 | 9.4 | 13.7 |
| | | | KD | 3.6 | 5.4 | 9.7 | 14.0 |
| | | | OTTER | **3.7** | **5.5** | **9.9** | **14.3** |
| CLIP (400M) | ResNet50 | CLIP | CLIP | 13.5 | 19.7 | 30.5 | 39.4 |
| | ViT-B/32 | Transformer | | 15.3 | 22.2 | 33.9 | 43.3 |

$\lambda = 0.15$ and the number of iterations to 5. For the KD baseline, we also use EMA teacher and set $\alpha = 0.5$. For the label-smoothing baseline, we set $\alpha = 0.9$, which yields better results than $\alpha = 0.5$.

On CC, we train the image-text models based on four different pretrained image encoders: ResNet-{50, 34} (He et al., 2016), FBNetV3-{A, C} (Wu et al., 2019b; Wan et al., 2020; Dai et al., 2020), and two pretrained text encoders: DeCLUTR-Sci-base (Giorgi et al., 2020) pretrained on S2ORC (Lo et al., 2020) and Sentence BERT (Reimers & Gurevych, 2019) pretrained on SNLI (Bowman et al., 2015) and MultiNLI (Williams et al., 2018). We also train ResNet50 + DeCLUTR-Sci-base on the (partial) WIT (Srinivasan et al., 2021) and the YFCC15M (subset of YFCC 100M) (Thomee et al., 2016) datasets. We report FH@K=1, 5, 10 on the test sets of both GOI and multi-labeled ImageNet 10K (Wu et al., 2019a). As shown in Table 2, over the 42 evaluations on 7 different dataset-architecture settings x 6 metrics, OTTER outperforms (32) or ties (2) all other baselines in 34 of them. Compared with CLIP's performance on the GOI test set, a ResNet50 trained by OTTER outperforms CLIP-RN50 by 2.6 pts FH@1 and by 6.6 pts FH@10. To further illustrate the significance of the performance gain, we show that a ResNet50 (25.6M params) trained with OTTER outperforms a Wide ResNet50x2 (68.4M params) trained with InfoNCE under the same setting.

For reference, to put OTTER in the context of traditional ZSL methods, we present FH@K results on zero-shot transfer to the ImageNet 21K+1K (Deng et al., 2009) dataset, which contains 21,841 classes in total. The result is reported in Table 3. With 400M image-text pairs, CLIP (Radford et al., 2021) vastly outperforms all other methods. ImageNet22K's classes contain many uncommon words, such as scientific names of animals or medical terms. While not directly comparable with traditional ZSL methods due to differences in datasets used and model architectures, OTTER is significantly better than previous ZSL methods, beating the previous SotA, HZSL(Liu et al., 2020), by 68% relatively.

## 4.2 VISUALIZING OTTER

In order to check if the image/text matching found by OTTER is sensible, we provide visualizations of OTTER's matching results. In Figure 3, we visualize the matching results on a small batch of 9 image-text pairs. We set $\alpha = 0.5$ for paired image-text samples, as shown in the diagonal elements in Figure 3. The off-diagonal elements are estimated by OTTER. Since the interpretation of the matching results are highly subjective, we leave the interpretation to readers.

Next, we use OTTER to process a larger batch of 512 image-text pairs. This is our batch size for training. We pick the top-8 largest off-diagonal pairs from the optimal tranport result and show them in Figure 4. As we can see, in a large batch, we can easily find unpaired images and captions that turn out to be good matches. InfoNCE will simply regard these pairs as negative examples and push them away from each other while OTTER can better handle this by treating them as semi-positive pairs.

## 4.3 IMPORTANCE OF SIMILARITY MATRIX AND EMA

In Equation 7, we design the similarity matrix $\mathbf{S}^v$ as the composition of image, text, and image-text similarity matrices. In Table 4, we show experiments to validate the effectiveness of this composition and the necessity of using EMA. There are various levels of performance drop when we don't use the image or text similarities, or when EMA is turned off. Note that our baseline hyper-parameters are **different** from Table 2, so the accuracy is also different. We compare different settings using FH@K=1 on the GOI test set. More in-depth ablation studies are shown in Appendix A.

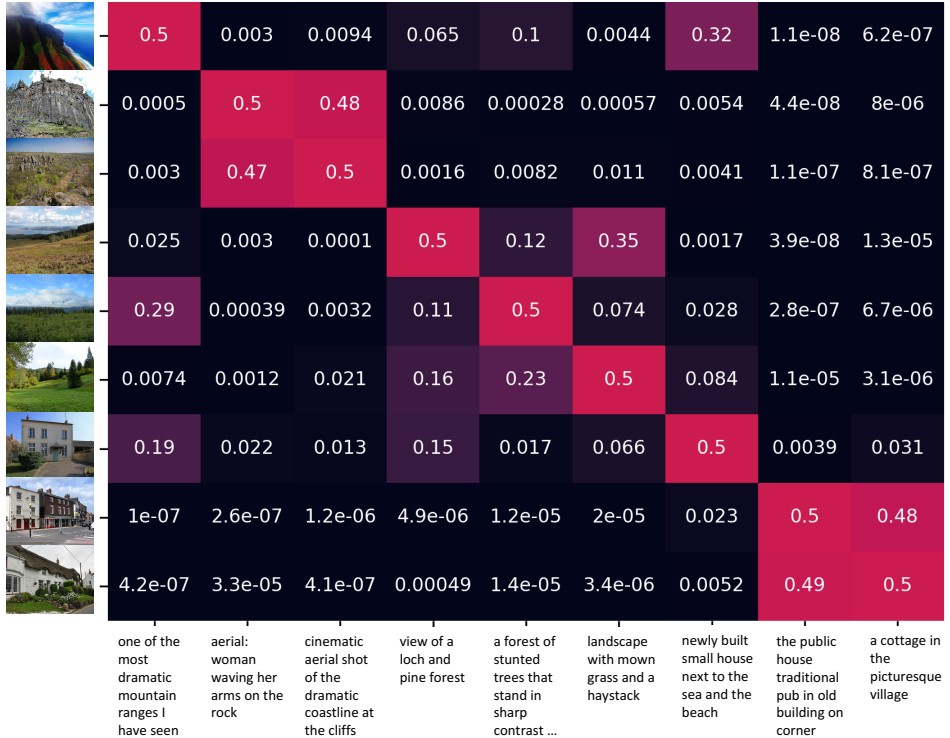

Figure 3: Visualization of OTTER's matching on a batch of 9 image/text pairs.

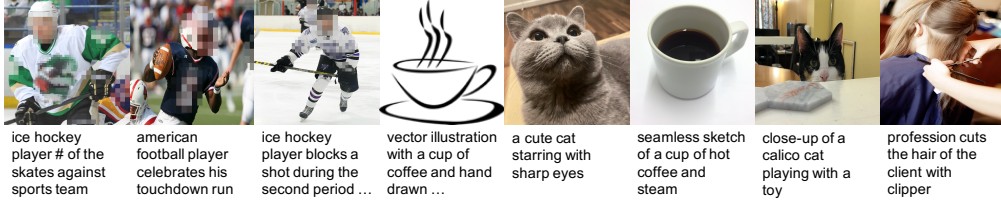

Figure 4: Visualization of top-8 image-text pairs matched by OTTER in a batch of 512 samples. These image/text pairs are regarded as negative samples by InfoNCE.

Table 4: Validation of Similarity Matrix and EMA.

| | $\alpha$ | $\gamma_v$ | $\gamma_t$ | EMA | $\lambda$ | #iter | batch | FH@K=1 |
|---|---|---|---|---|---|---|---|---|
| baseline | 0.5 | 1.0 | 1.0 | ✓ | 0.1 | 4 | 512 | 31.0 |
| similarity matrix | 0.5 | 0.0 | 1.0 | ✓ | 0.1 | 4 | 512 | 28.8 (↓ 2.2) |
| | 0.5 | 1.0 | 0.0 | ✓ | 0.1 | 4 | 512 | 27.8 (↓ 3.2) |
| | 0.5 | 0.0 | 0.0 | ✓ | 0.1 | 4 | 512 | 26.1 (↓ 4.9) |
| EMA | 0.5 | 1.0 | 1.0 | ✗ | 0.1 | 4 | 512 | 30.4 (↓ 0.6) |

## 5 CONCLUSION

Image-text datasets collected from the Internet are noisy, and the InfoNCE loss used by previous works such as CLIP fails to recognize the potential matches between unpaired images and captions in a batch. As a solution, OTTER extends the InfoNCE loss to consider the many-to-many relationship between unpaired images and texts by computing a pair-wise similarity matrix and using entropic optimal transport to solve for the off-diagonal matching probabilities. OTTER outperforms (32) or ties (2) all other baselines on Google Open Images and ImageNet 10K in 34 out of 42 comparisons. In future research, we want to test the effectiveness of OTTER on larger datasets, such as CLIP 400M.

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

# A   ABLATION STUDIES

In this section, we analyze the impact of hyper-parameters on the performance of OTTER in Table 5. Note that our baseline hyper-parameters are **different** from Table 2, so the accuracy is also different. We compare different settings using FH@K=1 on the GOI test set.

Table 5: Ablation studies. ResNet50 + DeCLUTR-Sci-base evaluated on GOI test set.

|  | $\alpha$ | $\gamma_v$ | $\gamma_t$ | EMA | $\lambda$ | #iter | batch | FH@K=1 |
|---|---|---|---|---|---|---|---|---|
| baseline | 0.5 | 1.0 | 1.0 | ✓ | 0.1 | 4 | 512 | 31.0 |
| $\alpha$ | 0.1 | 1.0 | 1.0 | ✓ | 0.1 | 4 | 512 | 29.9 (↓ 1.1) |
|  | 0.9 | 1.0 | 1.0 | ✓ | 0.1 | 4 | 512 | 28.4 (↓ 2.6) |
| similarity matrix | 0.5 | 0.0 | 1.0 | ✓ | 0.1 | 4 | 512 | 28.8 (↓ 2.2) |
|  | 0.5 | 1.0 | 0.0 | ✓ | 0.1 | 4 | 512 | 27.8 (↓ 3.2) |
|  | 0.5 | 0.0 | 0.0 | ✓ | 0.1 | 4 | 512 | 26.1 (↓ 4.9) |
| EMA | 0.5 | 1.0 | 1.0 | ✗ | 0.1 | 4 | 512 | 30.4 (↓ 0.6) |
| Sinkhorn | 0.5 | 1.0 | 1.0 | ✓ | 0.05 | 4 | 512 | 29.0 (↓ 2.0) |
|  | 0.5 | 1.0 | 1.0 | ✓ | 0.3 | 4 | 512 | 28.2 (↓ 2.8) |
|  | 0.5 | 1.0 | 1.0 | ✓ | 0.1 | 0 | 512 | 29.1 (↓ 1.9) |
|  | 0.5 | 1.0 | 1.0 | ✓ | 0.1 | 2 | 512 | 29.3 (↓ 1.7) |
|  | 0.5 | 1.0 | 1.0 | ✓ | 0.1 | 6 | 512 | 30.0 (↓ 1.0) |
| batch size | 0.5 | 1.0 | 1.0 | ✓ | 0.1 | 4 | 256 | 25.6 (↓ 5.4) |
|  | 0.5 | 1.0 | 1.0 | ✓ | 0.1 | 4 | 768 | 28.1 (↓ 2.9) |

**Confidence in the image-text pairs:** In Section 3.2, we define $\alpha = q_i^v(i)$ as the probability that the paired text caption is the correct match with the image. This reflects the confidence, or the noise level, in the ground truth pairs. We set $\alpha = 0.1, 0.5, 0.9$ in our experiment, and found that both lack of confidence (0.1) or over-confidence (0.9) can hurt the performance. Relatively, $\alpha = 0.9$ leads to worse performance, validating the necessity of mitigating label noise.

**Image-to-image, text-to-text similarity:** We included the image and text similarity when computing the pair-wise similarities for OT. The assumption is that samples with similar images or text captions are likely to share labels. To test this, we set $\gamma_v, \gamma_t$ to 0 in the experiments. We found that both image-to-image and text-to-text similarity are helpful. Relatively, text similarity seems to be more important than image similarity, as removing text similarity leads to a larger performance drop.

**Do we need EMA teacher?** In the default setting, we used the exponential moving average of the image/text encoders to compute the similarity estimation. We test the alternative option of using the image/text encoders themselves, and found that this leads to a small accuracy drop (0.6 points).

**Impact of optimal transport:** One key component of OTTER is to use optimal transport to match images with text captions within a batch. However, do we really need optimal transport? To compute the teacher distillation, a simple alternative is to use a Softmax function. As we discussed in Section 3.5, Softmax is equivalent to our Sinkhorn-Knopp implementation when we set the number of iterations to 0. So we validate the necessity of optimal transport by setting the #iteration to 0, 2, 4, 6. Experiments show that using Softmax (0 iteration of Sinkhorn) leads to the worst performance (-1.9). This validates the necessity of using optimal transport to ensure all images and texts within a batch are matchable. Besides, using 2 (fewer) and 6 (more) iterations also lead to accuracy drops (-1.7, -1.0). Using more iterations of Sinkhorn leads to a more converged solution to the optimal transport problem, but this does not seem to be positively correlated with better performance. We also explored the impact of entropy regularization controlled by $\lambda$ in Equation (6). Experiments show that the target distribution being too "hard" ($\lambda = 0.05$) or too "soft" ($\lambda = 0.30$) can hurt the performance.

**Batch size:** Previous works on contrastive learning show that a larger batch size raises the lower bound of mutual information (Hadsell et al., 2006) and leads to better performance (Chen et al., 2020b; He et al., 2020). However, for noisy image-text pairs, larger batch sizes can potentially bring more unpaired matches. We study the impact of batch sizes by setting it to $256, 512, 768$ in the experiments. We find that both smaller (256) and larger (768) batch sizes lead to worse performance. We hypothesize that batch size needs to be co-adapted with hyper-parameter settings of $\alpha$ and $\lambda$.

$\alpha$ estimates the noise level, which is positively correlated with the batch size. $\lambda$ yields different "softness" with different batch sizes. However, further investigation is to required to validate this.

## B    PSEUDOCODE FOR OTTER

---

**Algorithm 1:** PyTorch Pseudocode for OTTER

```
# fs, ft: student and teacher model.
# tpi, tpd: learnable inverse temperature.
# eta: a large constant, e.g., 100.
# alpha: loss coefficient.
# I_N: NxN identity matrix.
# xent: cross entropy function.

for img, txt in loader:
  # Regular InfoNCE loss
  emb_v, emb_t = fs(img, txt) # normalized embeddings.
  logits = emb_v @ emb_t.T
  prob_v = Softmax(logits * tpi) # normalize over t.
  prob_t = Softmax(logits.T * tpi) # normalize over v.
  L_infoNCE = xent(prob_v, I_N) + xent(prob_t, I_N)

  # Similarity estimation
  emb_v_t, emb_t_t = ft(img, txt).detach() # stop gradient.
  sim_vv, sim_tt = emb_v_t @ emb_v_t.T, emb_t_t @ emb_t_t.T
  sim_vt, sim_tv = emb_v_t @ emb_t_t.T, emb_t_t @ emb_v_t.T
  S_v = sim_vv + sim_tt + sim_vt - eta * I_N
  S_t = sim_tt + sim_vv + sim_tv - eta * I_N

  # Optimal Transport Distillation
  M_v = sinkhorn(S_v)
  M_t = sinkhorn(S_t)
  L_d = xent(prob_v, M_v) + xent(prob_t, M_t)

  # Final loss
  loss = alpha * L_infoNCE + (1-alpha) * L_d
  update(fs, ft, tpi, tpd)
```

---

**Algorithm 2:** PyTorch Pseudocode for Sinkhorn-Knopp

```
def sinkhorn(S, lambda=0.15, niter=5):
  T = exp(S / lambda)
  T = T / T.sum()
  N = T.shape[0]

  # iterative row/column normalization
  for _ in range(niter):
    T /= (T.sum(dim=1, keepdim=True) * N)  # row normalization
    T /= (T.sum(dim=0, keepdim=True) * N)  # column normalization

  # Note if niter=0, this is equivalent to Softmax
  return T /= T.sum(dim=1, keepdim=True)  # row normalization
```

---

## C    VARIANCE ANALYSIS

In our experiments, we noticed variance of experimental results with identical settings. To study this, we repeat the experiments in Table 2 with a ResNet50 image encoder and a DeCLUTR-Sci-base text encoder for 3 times each using different random seed to analyze the variance of the experiments. We noticed higher variance on GOI experiments. For example, for the FH@10 metric, the variance can be up to 1.88 pts. Note that the mean accuracy's gap between OTTER and baselines are all significantly larger than the standard deviation, indicating that the performance improvement of OTTER is not a

result of randomness. However, such high variance is worth noting and requires future investigation on how to reduce it.

Table 6: Flat hit @K on test sets of Google Open Images and ImageNet10K from Tencent-ML-Images.

| Data | Method | GOI FH@K (%) | | | IN10K FH@K (%) | | |
|---|---|---|---|---|---|---|---|
| | | 1 | 5 | 10 | 1 | 5 | 10 |
| CC (3M) | InfoNCE | $27.1 \pm 0.23$ | $56.1 \pm 0.92$ | $66.4 \pm 1.05$ | $10.9 \pm 0.38$ | $29.4 \pm 0.75$ | $40.5 \pm 0.76$ |
| | LS | $26.7 \pm 1.00$ | $55.9 \pm 1.31$ | $67.5 \pm 1.31$ | $10.1 \pm 0.67$ | $29.6 \pm 0.81$ | $39.8 \pm 1.03$ |
| | KD | $26.7 \pm 0.81$ | $55.3 \pm 1.67$ | $67.1 \pm 1.71$ | $10.0 \pm 0.75$ | $27.5 \pm 1.42$ | $38.5 \pm 1.14$ |
| | OTTER | $\mathbf{28.6 \pm 1.17}$ | $\mathbf{59.6 \pm 1.71}$ | $\mathbf{70.9 \pm 1.88}$ | $\mathbf{12.0 \pm 0.31}$ | $\mathbf{31.8 \pm 0.40}$ | $\mathbf{42.1 \pm 0.26}$ |

## D QUANTITATIVE ANALYSIS ON THE IMAGE-TEXT COMPOSITIONALITY ON CUB

ALIGN Jia et al. (2020) presents an interesting demonstration of the compositionality of image and text embeddings generated by language supervised vision models. On an image retrieval task, a query is formed by adding an image embedding vector to a text embedding vector. The returned image is expected to be similar to the image query and the text query. ALIGN demonstrates the compositionality by qualitatively showing several retrieval results, but does not provide any quantitative evaluations. In this paper, we design a preliminary benchmark based on the CUB dataset (Welinder et al. (2010)) to evaluate the image-text compositionality.

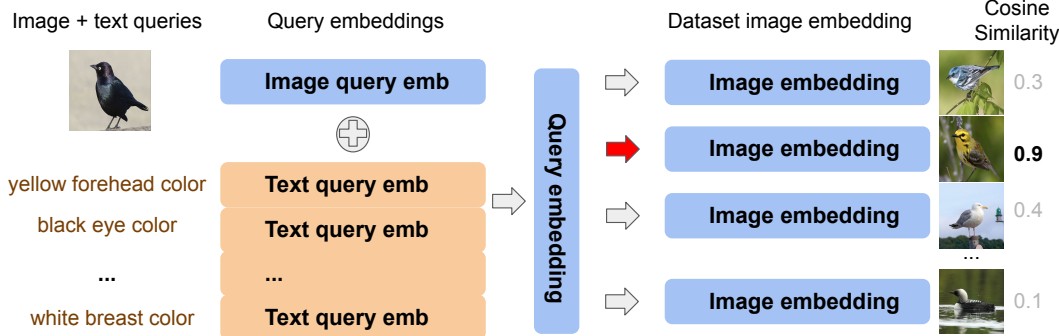

Figure 5: Illustration of the image + text -> image retrieval.

CUB (Welinder et al. (2010)) consists of 6033 bird images, and each image-$i$ is annotated with a set of bird attributes, which we denote as $A_i$. In this dataset, there are in total 288 unique attributes. Given an image and several CUB bird attributes in text, we generate image and text embeddings using our pretrained models and add the embeddings together to form a query embedding vector. We use the query vector to match images in CUB, and choose the nearest neighbor, based on cosine similarity, as the retrieved image. This process is illustrated in Figure 5. To evaluate the retrieval quality, we compare the overlap between the retrieved image's attributes with the image-text query's attributes.

We now describe how we generate text queries in addition to an image query. Randomly adding text attributes to image queries may result in unmatchable queries. To ensure that the added text query is sensible and the combined image-text query can be matched to an image in the dataset, we obtain a text query with the following approach: We first select a pair of images, denoted as image-$i$ and image-$j$. We use image-$i$ as the image query, and let $Q^v = A_i$ be the image query's attribute set. Then, to form the text query, we compare the differences of image-$j$ and $i$, and let $Q^t = A_j - A_i$ be the text query's attribute set. The combined image-text query should contain attributes $Q = Q^v \bigcup Q^t$. We show an example of image query, text query, and combined query in Figure 6. Following this process, we generate 1,000,000 image-text queries from randomly sampled image pairs from CUB where each image pair shares at least 10 common attributes. The image-text queries used in our experiment are provided in the attached file `image_text_query_list.txt` in the supplementary material.

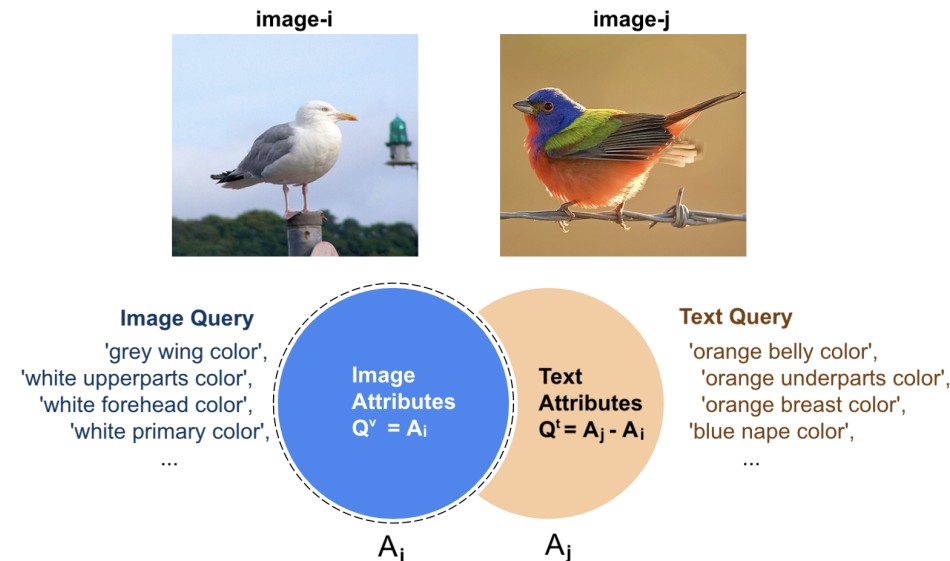

Figure 6: Image and text query example.

Table 7: Quantitative Vision-Language Compositionality Benchmark. OR represents Overlapping Rate, IOR represents Image Overlapping Rate, and TOR represents Text Overlapping Rate.

| Model | Image Encoder | Text Encoder | Method | OR (%) | IOR (%) | TOR (%) |
|---|---|---|---|---|---|---|
| Baseline | - | - | - | 27.7 | 31.3 | 23.2 |
| CLIP | ResNet50 | CLIP Transformer | InfoNCE | 35.7 | 36.2 | 34.5 |
| Ours | ResNet50 | DeCLUTR -Sci-base | InfoNCE | 34.5 | 33.6 | **36.8** |
| | | | LS | 34.2 | 33.6 | 35.8 |
| | | | KD | 33.2 | 32.9 | 34.4 |
| | | | OTTER | **34.7** | **33.9** | 36.7 |

Our evaluation measures the overlap between the retrieved image's attribute set $R_k$ with the image-text query set $Q_k$, image query set $Q_k^v$, text query set $Q_k^t$. Specifically, we compute: 1) the attributes overlapping rate (OR) $\frac{1}{N} \sum_{k=1}^{N} \frac{|Q_k \cap R_k|}{|Q_k|}$. This measures the retrieval quality for the combined image-text query. 2) the average image attributes overlapping rate (IOR) $\frac{1}{N} \sum_{k=1}^{N} \frac{|Q_k^v \cap R_k|}{|Q_k^v|}$. This evaluates if the returned image hits/misses attributes in the image query. 3) the average text attributes overlapping rate (TOR) $\frac{1}{N} \sum_{k=1}^{N} \frac{|Q_k^t \cap R_k|}{|Q_k^t|}$. This evaluates if the retrieved image hits/misses attributes of text queries. An ideal match should achieve higher scores in OR, IOR, and TOR simultaneously.

We report our results comparing with CLIP and other baselines in Table 7. First, we report the measurement of a random baseline. With this random baseline, for any image-text query, we return a random image from the dataset. We can see that this gives a non-trivial OR (27.7%), IOR (31.3%), and TOR (23.2%). This is not surprising because images in CUB have many overalapping attributes. However, although both CLIP and our models are never directly trained on CUB nor trained to predict bird attributes, their overlapping rates are significantly higher than the random baseline. CLIP achieves comparable accuracy with our models, with 1% higher OR, 2.3% higher IOR, and -2.2% lower TOR. Among different training methods, OTTER outperforms all baselines in OR and IOR, and achieves slightly worse (-0.1%) TOR than the InfoNCE baseline. Also note that our models achieve similar IOR and TOR, which shows both image and text queries are considered. This demonstrates good compositionality.

In Figure 7, we show qualitative results of our model trained by OTTER. For example, in the first row, we add *yellow forehead color*, *black eye color*, *yellow upper-parts color* and *yellow breast*

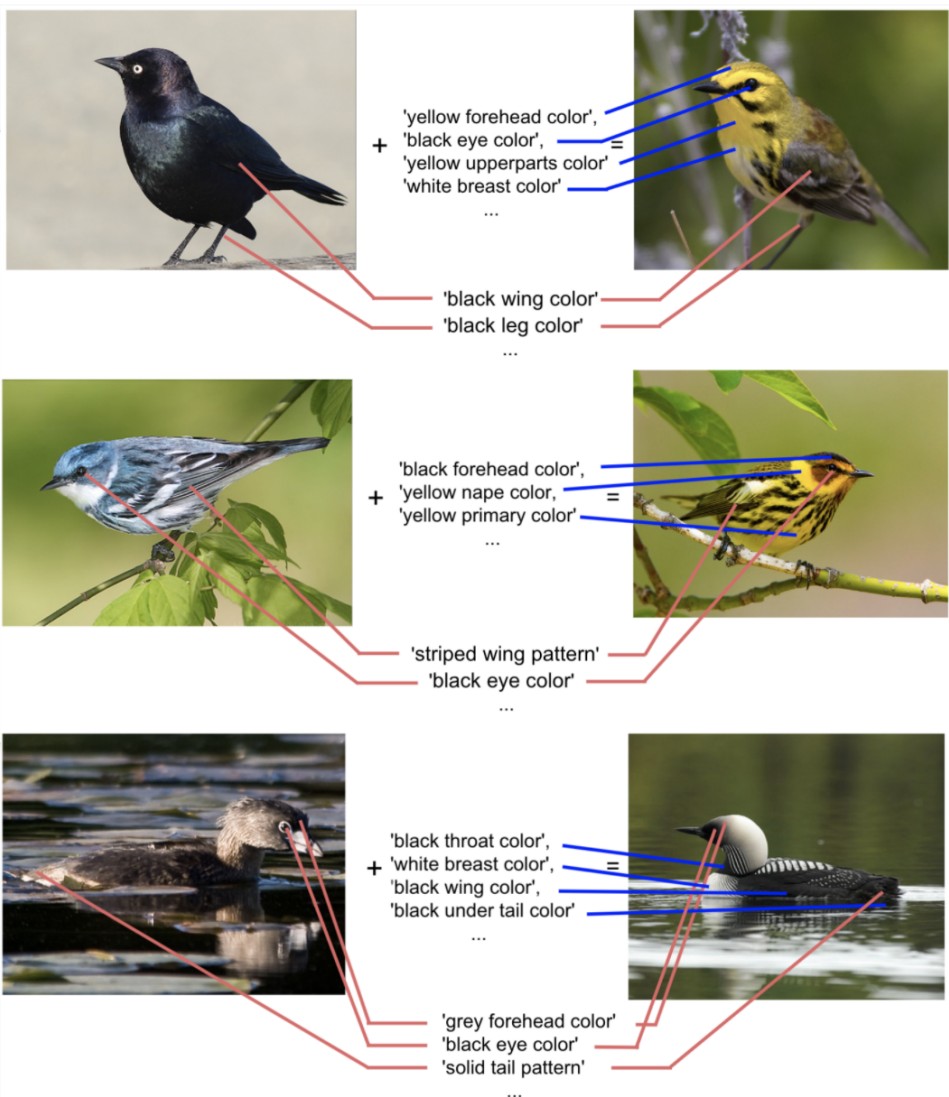

Figure 7: Qualitative results of text + image retrieval. Blue lines indicate texts attributes, and red lines indicate image attributes. For the demonstration purpose, we only label parts of the attributes that are easy to recognize.

*color* as text attributes to an image of a black bird. The retrieved bird image contains attributes from both image and text queries – it has *yellow body colors* while maintains *black leg* and *black wing* colors. From both quantitative and qualitative results, our model demonstrates good image-text compositionality.

## E    IMAGE ATTRIBUTIONS

1. Figure 1, Paul Bica, Coast of Kauai, Hawaii, CC BY 2.0
2. Figure 1, James St. John, Columnar-jointed rhyolitic obsidian lava flow, CC BY 2.0
3. Figure 1, Mordaka, QK9A1397, CC BY-SA 4.0
4. Figure 1, Alan Reid, Heathery moor on the flank of Stone Saul, CC BY-SA 2.0
5. Figure 1, Jimmy Emmerson, The Tormented Valley, CC BY-NC-ND 2.0
6. Figure 1, Roman Boed from The Netherlands, Black Forest- Meadow (10561897306), CC BY 2.0
7. Figure 1, Daniel Clerc / CC-BY-SA-3.0, 2013 bois herpin 013, CC BY-SA 3.0
8. Figure 1, Dave Bevis, 22 and 24 High Street, Newcastle-under-Lyme, CC BY-SA 2.0
9. Figure 1, Nilfanion, Thatched cottages in Coverack (8379), CC BY-SA 4.0
10. Figure 4, TheAHL, Chuck Kobasew (cropped), CC BY 2.0
11. Figure 4, Mark Mauno, Jasper Fitzi, CC BY 2.0
12. Figure 4, Famartin, 2020-04-27 18 55 23 A Calico cat looking for food in a kitchen in the Franklin Farm section of Oak Hill, Fairfax County, Virginia, CC BY-SA 4.0
13. Figure 4, Beth, Haircut-4, CC BY 2.0

