# OpenReview forum: "Data Efficient Language-Supervised Zero-Shot Recognition with Optimal Transport Distillation"
_ICLR.cc/2022/Conference — ICLR 2022 Poster_

### Official Review · Reviewer_ayPf · 2021-11-01

**Correctness:** 3
**Technical Novelty And Significance:** 3
**Empirical Novelty And Significance:** 2
**Recommendation:** 5
**Confidence:** 4

**Main Review:**

Strengths
1)	The authors provide an interesting and solid idea based on CLIP.
2)	The presentation is very good and easy to follow. The authors clearly present their ideas and describe the technical details.
3)	The ablation study and visualization analysis of the experimental results are sufficient. The visualization of OTTER’s matching illustrates its effectiveness in handling many-to-many relationships.

Weakness
While the ablation study and visualization analysis are done, the key evaluations are missing. The details can be found as follows.

The experiment results in Tables 1 & 2 cannot plausibly prove OTTER is more effective than CLIP. The larger dataset may relate to more many-to-many relationships when training the model. For example, in the visualization in Figure 2, the maximum many-to-many relationship of a sample is 3. However, the dataset with 400M may contain too many many-to-many relationships like 7 or 8 (maybe more, the data scale is about 100 times the used datasets in this paper). In this case, OTTER may be degraded due to the multiple learning targets. In contrast, focusing on a solid target and ignoring the rest of potential targets, CLIP may still infer the generalized information provided by many-to-many relationships due to the wide range of data collection. Therefore, it is necessary to compare OTTER and CLIP on the same-scaled datasets. The authors provide OTTER using hard labels (InfoNCE) as a baseline, but ZSL methods are sensitive to hyper-parameters and training sets. I doubt that InfoNCE can represent the best performance of CLIP trained on CC (3M) and WIT(5M). Regarding that CLIP does not release the 400M dataset (mentioned in this paper by the authors), the authors may not be able to train OTTER on the 400M dataset. The provided experiments are more like baselines for self-evaluation other than state-of-the-art performance. Without the experimental results using the same training settings, I do not think the authors are able to prove the superiority of OTTER over CLIP fully.


**Summary Of The Paper:**

This paper proposes an enhanced model Optimal TransporT distillation for Efficient zero-shot Recognition (OTTER) based on Contrastive Language–Image Pre-training (CLIP). The authors present the potential many-to-many visual to text relationships that CLIP cannot handle. The authors proposed to model the many-to-many relationships as optimal transfer, which portrays the probability relationship based on two intuitions: 1) all the images and texts are equally matchable and have equal matching probabilities; 2) the matching probability from image to caption should depend on their similarity estimation. The authors further use knowledge distillation to enhance the learning of similarity matrices.

**Summary Of The Review:**

According to my main review, I believe this paper still has room to improve

---

> ### Author Response · Authors · 2021-11-23
> **Response to Reviewer ayPf**
>
> Thank you for your review and for noting that we provide an interesting and solid idea and our presentation is easy to follow.
>
> **Q. “Tables 1 & 2 cannot plausibly prove OTTER is more effective than CLIP.”**
>
> First, we would like to respectfully point out that CLIP models trained with 400M image-text pairs is not a baseline of OTTER. The datasets, training procedures, and model architectures etc. are all different, thus making CLIP and OTTER not directly comparable. However, InfoNCE is the core algorithm used by CLIP, and is the state-of-the-art technique in language-supervised zero-shot learning. Table 1 provides a direct comparison between InfoNCE and OTTER trained on identical datasets with identical model architectures and training procedures.
>
> The comparison with the traditional ZSL methods in table 2 is to provide readers a reference and put OTTER in the context of traditional ZSL methods (with lower performance than us) and modern ones such as CLIP (with stronger performance than us). Please also see our response to Reviewer qBzH (“Comparison with other ZSL methods in table 2 is unfair”).
>
> **Q. Does OTTER still outperform InfoNCE on larger datasets?**
>
> Yes, please see response to Reviewer DJW4 (“Does OTTER work on larger-scale datasets?”).
>
> **Q. How prevalent is the noisy matching problem? Does a larger dataset mean noisier samples?**
>
> Please see response to Reviewer qBzH (“How prevalent is the problem of many-to-many image and text matching in the datasets?”).

---

> > ### Comment · Reviewer_ayPf · 2021-11-29
> > **Response to authors**
> >
> > Thanks for the authors' response and extra experiments. Some of my concerns are addressed.
> >
> > Regarding the CLIP, as the authors claimed in the paper that OTTER is more efficient, as I said in my comment --  ``it is necessary to compare OTTER and CLIP on the same-scaled datasets.''. I think authors should compare the performance with CLIP in the same dataset as the major motivation of this paper is that OTTER is data-hungry and inefficient.
> >
> > Therefore, I would like to keep my score.

---

> > > ### Author Response · Authors · 2021-11-29
> > > **Follow-up to Reviewer ayPf**
> > >
> > > Thanks again for your comments. I would like to reiterate that CLIP is not our baseline, and in the experiments section, it's very clear that OTTER is able to outperform InfoNCE under the exact same training setting, on multiple datasets including YFCC 15M. Please see response to Reviewer DJW4 (“Does OTTER work on larger-scale datasets?”).
> > >
> > > It is also difficult for us to directly quote numbers from CLIP, since Open AI has not released their models trained on YFCC 15M. Our InfoNCE baseline is our best effort to replicate CLIP's performance, given the computational constraints. We also have quoted the performance of CLIP pertained on the much larger 400M dataset in the paper, for reference. A CLIP model pertained on YFCC15M will likely achieve a lower number than the numbers we already provided in the paper.

---

### Official Review · Reviewer_pk8X · 2021-11-02

**Correctness:** 3
**Technical Novelty And Significance:** 3
**Empirical Novelty And Significance:** 4
**Recommendation:** 6
**Confidence:** 4

**Main Review:**

---
Strengths:

1. The work has a clear hypothesis and is well-motivated. Many-to-many relationships / soft correspondences are generally important for vision-and-language learning since there is never really one unique and perfect correspondence between an image and text. Additionally, improving data efficiency of pre-training vision-and-language models can be beneficial to the community as many recent works rely on enormous amounts of image-text data. Others would likely upon this work in a similar direction.

2. The proposed method addresses the issue of many-to-many relationships in a technically valid way, and doesn't increase the complexity of the general approach significantly

3. The authors present experiments with a good set of baselines (LS, KD, InfoNCE) as alternative methods and ablation studies on each component of their approach

4. The experiments show that the proposed approach consistently outperforms alternative approaches (baselines)


---
Weaknesses:


1. Although the proposed approach doesn't increase the complexity significantly, it does introduce a number of new hyperparameters – many of which the models seem highly sensitive to (Table 3).
Intuitively, different datasets might have more or less similarities between samples, noise level, etc. - therefore the hyperparameters that work well on CC & WIT, which are of relatively high quality, might not be very transferable to more noisy datasets (like the one used in CLIP, for example).
The ablation studies indicate that, e.g., batch size has a very significant impact on the performance, and intuitively the best choice should depend on the particular dataset since, for a given batch size, the average number of similar samples might vary greatly among different datasets.

2. The comparison with ZSL methods trained on ImageNet1K (Table 2) is misleading. These ZSL methods are trained on ImageNet1K and evaluated on ImageNet21K because of a relatively clear separation between seen and unseen (novel) classes. However, the proposed approach is trained on the CC dataset, which likely has a substantial overlap of concepts with the test set (ImageNet21K). Therefore, the comparison of the prior works and the proposed approach and claims of improving state-of-the-art are somewhat misleading because training on CC means that the model has effectively seen images + names (likely in captions) those classes that are supposed to be "unseen". Even though all works can be said to do zero-shot, the prior methods trained on ImageNet1K do zero-shot with respect to novel classes/concepts, while the proposed method & CLIP model do something more of zero-shot with respect to a dataset (but having possibly seen at least some of the concepts/classes already). Thus, making a side-by-side comparison without discussing this difference can be misleading.

3. The problem of noise in datasets is discussed in the context of more than one possible matching for a given text/image. But what about the image-text pairs not being "accurate" - e.g. text not relevant, or not descriptive enough?


---
Other:

- Question: Why do both $M^V$ and $M^T$ need to be computed (for the OTTER variant)? Shouldn't they be the same (at least for a larger number of Sinkhorn-Knopp iterations)?

**Summary Of The Paper:**

The paper proposes an image-text training method similar to CLIP. However, in training, instead of using hard targets for image-text pairs, it relies on soft image-text correspondences based on pairwise similarities and optimal transport.

As image-text pairs can possibly have more than one positive match in the training batch, the authors motivate that the usage of soft targets is more appropriate because it better reflects the possible many-to-many relationship between the image and text samples.

The authors claim that such an approach leads to better data efficiency as it achieves higher performance on CC and WIT datasets, which are significantly smaller than the one used by CLIP.

**Summary Of The Review:**

The work introduces a valid approach for considering many-to-many correspondences in image-text datasets which is an important problem for vision-and-language learning.

Although the presented results show consistent improvements over alternative approaches, the approach does introduce a few new hyperparameters. Intuitively they might need to be tuned carefully for different datasets to be able to observe any improvement over simpler approaches. The ablation studies do suggest that within a given dataset the sensitivity to some of those hyperparameters is high.

Additionally, the work makes a side-by-side comparison with ZSL prior works, which, unlike this paper, do control the separation between seen and unseen classes/concepts in their training/test data. Such comparison and claims of exceeding ZSL their performance and reaching a new state-of-the-art might lead to misleading conclusions.

**Update (after the authors' response):**

Raised the recommendation to _6: marginally above the acceptance threshold_

---

> ### Author Response · Authors · 2021-11-23
> **Response to Reviewer pk8X**
>
> Thank you for your review and for recognizing that our paper is well motivated and our method addresses the problem in a technically valid way.
>
> **Q. There are many hyperparameters introduced by OTTER. How should we set alpha based on batch size and dataset?**
>
> Please see response above to Reviewer DJW4 (“Analysis of the hyperparameters”).  In the response to Reviewer qBzH (“How prevalent is the problem of many-to-many image and text matching in the datasets?”) above, we show that we could use a pretrained CLIP model to estimate **Alpha** for a given dataset and batch size.
>
>
> **Q. A side-by-side comparison with traditional ZSL methods is misleading.**
>
> Please see response to Reviewer qBzH (“Comparison with other ZSL methods in table 2 is unfair”).
>
> **Q. What about the case where the default image-text pairs are not "accurate" enough?**
>
> Besides the many-to-many matching problem, it is also common that the default caption may not be accurate or descriptive enough, less so in filtered/curated datasets such as CC3M, and more so in datasets like YFCC15M with longer and noisier captions. OTTER is designed to handle these situations too, by assigning a lower-than-one probability for the default pairing and identifying the potential matches in the non-paired captions in the batch. Please also see response to Reviewer qBzH (“How prevalent is the problem of many-to-many image and text matching in the datasets?”)
>
> **Q. Why do M^V and M^T need to be computed separately?**
>
> If Gamma_v and Gamma_t in Eq 7. are both set to 1 (default case for OTTER), then M^V and M^T are just transposes of each other. In the code implementation, however, embeddings computed from different GPUs are concat_all_gathered without gradients, so M^V and M^T needs to be computed separately, as shown in the pseudocode.

---

> > ### Comment · Reviewer_pk8X · 2021-11-29
> > **Response to the authors**
> >
> > I would like to thank the authors for their extensive response!
> >
> > After reading the response **I am willing to raise my evaluation to "6. Marginally above the acceptance threshold".**
> >
> > The authors to some extent addressed most of my concerns:
> > - Extra hyperparameters: Convincing response that most of the hyperparameters are simple to set and don't require much tuning. Also, I appreciate the extra results showing match statistics over different batch sizes & datasets which suggest that a reasonable choice of alpha value is possible
> > - Misleading comparison with ZSL methods: rephrased discussion on the comparison with ZSL methods should be less confusing to the readers, although it still does not explicitly discuss the lack of separation between seen and novel classes, only more general "difference in datasets used"
> >
> > Additionally, I appreciate the extra experiments on YFCC 15M and the comparison with wide resnet. They do help to understand the significance of the results/improvements.
> > Finally, the statistics of on/off-diagonal matching do provide extra insight into the importance of considering matching other than one-to-one.

---

### Official Review · Reviewer_DJW4 · 2021-11-02

**Correctness:** 3
**Technical Novelty And Significance:** 3
**Empirical Novelty And Significance:** 2
**Recommendation:** 6
**Confidence:** 5

**Details Of Ethics Concerns:**

No ethic issues spotted in the paper.


**Main Review:**

Strengths:
+ The paper is very-well written and easy to follow. In particular, the approach section is clearly written and presented in a good pedagogical manner.
+ Overall, IMO the proposed approach solves the missing match problem in language-vision contrastive pretraining in a sensible way.
+ The results seem to suggest that the proposed approach is overall a good fix to existing language-vision pretraining methods, as it generally provides a gain (in certain cases bigger than others) across different target datasets and choice of encoder architectures.
+ The authors have made good efforts to ablate different components of their approach.
+ The authors have promised to release their code for training and evaluation.

Specific things I like in the paper:
+ I like Section 3.5 as it helps clearly distinguish the proposed method to related methods with term swapping.
+ I like the visualization in Figure 3, which gives me a sense of what kind of missing matchings are mined.

Weaknesses:
- One issue I have for the approach is that, it's not clear to me why self-distillation is necessarily beneficial? Since the authors use the original encoder (or their momentum-updated counterparts) as teachers, what extra information does the distillation bring in? At a conceptual level, the only extra source of information is the introduction of text-text and visual-visual similarities in Eq. 7. But from the results (i.e., 'running SK for 0 iterations' vs the full OTTER method), it seems to suggest that's not the only reason for the improved accuracy. It would be good if the authors could have a summary on where they think all the gains come from.
- The authors have motivated their approach by giving the example in Figure 1 to show that, there are many missed matches between images and texts. However, there lacks a quantitative study showing how prevalent this problem is. Although one can argue from the improved results, it would be more convincing to have a ballpark of how many missed matches there are in a batch.
- The proposed method has many parameters (alpha gamma_v gamma_t, eta, lambda), there lacks a detailed sensitivity analysis for them (Table 3 only does coarse ablation that mostly turn on or off the term).
- Page 5, "we train on the two publicly available datasets, Conceptual Captions 3M (CC) (Sharma et al., 2018), and Wikipedia-base Image-Text Dataset (WIT)", it would be helpful to train on a larger and more noisier dataset like Conceptual Captions 12M to see how the proposed method scales (my impression is that the benefit of the proposed method should be more significant due to higher noise level there).
- In Table 1, for three image/text encoder pairs "FBNetV3-A/DeCLUTR-Sci-base", "FBNetV3-C/DeCLUTR-Sci-base" and "ResNet50/Sentence-BERT-base", OTTER only has very marginal gain over KD, especially on IN10k, any insights on why this is case?
- In Table 2, for OTTER, why not use ResNet50 as the image encoder to enable a more fair comparison? Again, for the "FBNetV3-C/DeCLUTR-Sci-base" setting, the gain is quite marginal between OTTER and KD.

Questions/confusions:
- Page 3, "Directly minimizing Wasserstein loss between image/text embeddings in our case will lead to collapsed representations", any insight why this might be happening?
- Page 3, "tau is a (trainable) temperature parameter", is there evidence showing the benefit of learning this parameter, instead handpicking one using validation? Also, what value does tau converge to?
- Table 3, it would be good to have results for setting lambda to 0 (no entropy regularization).
- Page 6, "with a total batch size of 512 (64 per GPU) for 10 epochs", how long does it take to train for 10 epochs?


**Summary Of The Paper:**

This paper tackles the problem of zero-shot image recognition through language-supervised pretraining. The proposed method trains their visual-language encoder pair with a contrastive objective (similar to CLIP) on image-text pairs. The main problem the authors saw and aim to mitigate from previous approaches like CLIP is the existence of missing-matches between a batch of image-text pairs for training. The authors argue that, in a batch, one image can be matched to multiple texts and vice versa. They propose to use Optimal Transport optimization to solve for a soft image-text matching to replace the hard single-to-single matching and serve as the pseudo ground-truth for supervision.

Then, the pretrained models visual and language encoders are taken to several target datasets (Google Open Images, ImageNet 10K, and ImageNet 21K+1K) for zero-shot recognition (text encoders are used to generate classifiers on the fly for labels in the target datasets). The proposed method demonstrated good results compared to several previous methods and also a set of variants for ablation studies.


**Summary Of The Review:**

Overall, I like the idea proposed in the paper to use OT to mine missing matches for language-vision pretraining. The paper is clearly written and makes intuitive sense for most part -- I'm overall positive on accepting this paper but still like to hear the authors address my concerns listed above.

---

> ### Author Response · Authors · 2021-11-23
> **Response to reviewer DJW4**
>
> Thank you for your review and for recognizing that our approach solves the mismatching problem in vision and language in a sensible way.
>
> **Q: Does OTTER work on larger-scale datasets?**
>
> Yes, we provide results of training both InfoNCE and OTTER on YFCC 15M using ResNet50 and Declutr-sci-base, to show that OTTER also works on a relatively large-scale dataset.
>
> | Dataset  | Method  | GOI F@K=1 | IN10K F@K=1 | IN 1K Acc | IN 21K Acc |
> |----------|---------|-----------|-------------|-----------|------------|
> | YFCC 15M | InfoNCE | 19.04     | 8.96        | 37.78     | 5.57       |
> | YFCC 15M | OTTER   | 20.59     | 9.29        |  38.37    | 5.66       |
>
> **Q: Why is self-distillation beneficial, and what information does it bring in? Where do all the gains on OTTER come from?**
>
> Self-distillation with soft-targets and ema has been proved to be an effective way of representation learning in many prior works, such as [1, 2]. OTTER uses soft targets estimated from the model output itself, and hence uses self-distillation, yet it’s not where the gains come from.
>
> The main performance gain of OTTER comes from three parts:
> - First, OTTER recognizes the noisy image-text matching problem and provides a problem formulation that extends InfoNCE and appropriately estimates off-diagonal probabilities. This issue is addressed by both KD and OTTER.
>
> - Second, the design of the similarity matrix takes into consideration image-image, text-text similarities, which is extra information not considered by InfoNCE. This is only addressed in OTTER but not in KD.
>
> - Third, by using optimal transport, we enforce a separate constraint not present in KD, i.e. all images and texts are equally important and should be matched with equal probabilities, as shown in Eq. 4 and Eq. 5. To put it differently, when estimating the image-to-text probability, this would prevent all the images from matching to only a few text captions, because all the texts are equally important. The same goes for the text-to-image case. This is also only addressed in OTTER but not in KD.
>
> We believe that the performance gain of OTTER against KD is attributed to the second and third parts.
>
> Ref:
>
> [1] Momentum Contrast for Unsupervised Visual Representation Learning
>
> [2] Emerging Properties in Self-Supervised Vision Transformers
>
>
> **Q: Quantitative study of the prevalence of the noisy image-text matching problem.**
>
> Please see our response to Reviewer qBzH (“How prevalent is the problem of many-to-many image and text matching in the datasets?”) above.
>
> **Q: Analysis of the hyperparameters.**
>
> We provide a comprehensive ablation study to 1) justify some key design decisions of OTTER and 2) show the impact of hyper-parameters on OTTER’s performance. However, we realize this seems to have created a confusion that OTTER has too many hyper-parameters and needs intensive hyper-parameter tuning. This is not true.
>
> In the paper, we report ablation studies on 7 hyper-parameters. In fact, only 2 hyper-parameters are needed by OTTER. Specifically, among the hyper-parameters reported in the paper,
> - **Gamma_v** and **Gamma_t** in Eq 7. should always be set to 1.0. We set them to 0 in the ablation study to show the effectiveness of considering image-image and text-text similarities.
> - **Eta** in Eq 7. should be set to a large number (say 10^6) to make sure the diagonal matching probability becomes zero).
> - **#iter** is how many iterations the SInkhorn-Knopp algorithm runs. It can be set to a reasonable number above 5, so that the sinkhorn converges (usually pretty fast). The softness of the probability output can be controlled by lambda.
>
> The only two hyper-parameters needed are:
> - **Alpha** is the prior probability given to the default image-text pairing. In the dataset analysis presented above, the normalized on-diagonal matching probability estimated by a pretrained CLIP VIT-B/32 on CC is 56.5%. We can set alpha close to this value, and indeed, using \alpha=0.5 yields the best result in the ablation studies.
> - **Lambda** controls the softness of the target probability output.
>
> Tuning these two hyper-parameters only requires modest compute cost.

---

> > ### Comment · Reviewer_DJW4 · 2021-11-26
> > **Thanks for the author's response**
> >
> > I thank the authors for their clarification, it explained away many of my concerns.
> > One followup question regarding "Quantitative study of the prevalence of the noisy image-text matching problem": since the average values for the off-diagonal elements are close to 0, while the max value doesn't tell us much about distribution, would it be possible to generate a histogram to show the distribution of the off-diagonal values?

---

> > > ### Author Response · Authors · 2021-11-29
> > > **Follow-up to Reviewer DJW4**
> > >
> > > Thank you for providing such valuable feedbacks and we are glad that our answers clarified some concerns. Here is the [Google Sheets link](https://docs.google.com/spreadsheets/d/1QhEUUh4w58G2I2oj9YD4TlVpPGWDdEw8_E3W3jSH-Vs/edit?usp=sharing) to the off-diagonal score histograms for both CC and YFCC. While it may look heavily skewed to the right, it's clear that high off-diagonal scores appear quite often in a 512x512 batch.
> > >
> > > In addition, in a 512x512 batch on YFCC, there are, on average, 387 off-diagonal scores > 0.1 and 213 off-diagonal scores > 0.2. On CC, there are 465 off-diagonal score > 0.1 and 244 > 0.2.

---

> > > > ### Comment · Reviewer_DJW4 · 2021-11-30
> > > > **Re: Follow-up to Reviewer DJW4**
> > > >
> > > > Thanks for providing the histogram! This is very helpful to understand the prevalence of the mismatch problem. I suggest adding these info into the revised paper.

---

> > > > > ### Author Response · Authors · 2021-11-30
> > > > > **Thanks for the comments**
> > > > >
> > > > > We really appreciate the detailed comments and we will make sure to include the suggested revisions and graphs into the paper.  We would also like to gently remind that today is the last day to update scores if you intend to do so, and we are happy to answer any other questions or concerns.

---

> ### Author Response · Authors · 2021-11-23
> **Response to reviewer DJW4 (part 2)**
>
> **Q: Concerns over performance gain in table 1**
>
> In table 1, KD also incorporates an estimation of off-diagonal probabilities, which partially solves the noisy matching problem. The differences between OTTER and KD are only the similarity matrix design and optimal transport constraint, and thus a smaller performance gain is expected.
>
> On IN10K, the result numbers are generally lower and thus the performance gain may seem smaller. For an extra perspective on performance gain, we have trained a wide_ResNet50x2+Declutr-Sci-base with InfoNCE. Please see response to Reviewer qBzH (“ Are the improvements shown in table 1 significant?”).
>
> **Q: Concerns over performance gain in table 2**
>
> In table 2, the comparison with the traditional ZSL methods is for reference, to put OTTER in the context of traditional ZSL. We don’t claim that our method is directly better than the traditional methods, but having the data provides another perspective of the performance of OTTER.
>
> ### Confusions:
> **Q. "Directly minimizing Wasserstein loss between image/text embeddings in our case will lead to collapsed representations". Why?**
>
> In [1], the Wasserstein loss works under representation distillation because the teacher model is fixed. In our setting, however, the teacher model is a delayed version of the student model. A trivial solution to minimize the Wasserstein loss is that the output embedding stays constant regardless of the input. This is not likely to happen with a fixed teacher, but is very common in our setting [2, 3].
>
> If the model simply learns to output a collapsed representation, e.g. a zero vector, then both the teacher and student embeddings are zero, and the Wasserstein loss also becomes zero.
>
> Ref:
>
> [1] Wasserstein Contrastive Representation Distillation
>
> [2] Momentum Contrast for Unsupervised Visual Representation Learning
>
> [3] Bootstrap your own latent: A new approach to self-supervised Learning
>
> **Q. Learning vs hand-picking Tau? What value does Tau converge to?**
>
> In our experiments, Tau becomes relatively stable fairly quickly and converges to around 60. So empirically, we can set Tau to 60 directly.
>
> **Q. Table 3, what if we set lambda to 0?**
>
> If we look at Eq 4., setting lambda to 0 would mean that we don’t care about the entropy of the target probability. Thus, we will get a sparse matrix that satisfies the constraints in Eq 5., i.e. for each row and column, there is one entry with a 1, the rest are all 0s. Experimentally, running the sinkhorn algorithm with a lambda of 0 is not numerically stable (algorithm 2, page 14).
>
> **Q. How long does it take to train "with a total batch size of 512 (64 per GPU) for 10 epochs"?**
>
> To train a RN50 + Declutr-Sci-base on CC3M for 10 epochs with 8 V-100 GPUs and a batch size of 512, it takes around 12 hours to train InfoNCE and 16 hours to train OTTER. Inference speed is exactly the same.

---

### Official Review · Reviewer_qBzH · 2021-11-02

**Correctness:** 3
**Technical Novelty And Significance:** 4
**Empirical Novelty And Significance:** 2
**Recommendation:** 6
**Confidence:** 4

**Main Review:**

Strengths:

1. The authors showcase a fundamental problem of InfoNCE by providing interesting examples in Figure 1 and Figure 3. This is a valuable insight for the community, and may encourage future work to make more realistic assumptions about image-caption datasets.
2. The proposed method is simple and elegantly formulated. It is versatile and can be applied to many vision-language tasks besides zero-shot recognition. There are hyperparameters alpha and lambda that control how much we trust the image-caption pairs, and how much we trust the similarity estimations made by the teacher model.
3. The paper is well-written and easy to follow.
4. The experiments show consistent improvements in various training and evaluation settings, on difficult and realistic classification tasks.

Weaknesses:

1. Although the improvements shown in Table 1 are consistent, they are not significant. Hence, using OTTER may not be worth the extra computational cost.
2. The comparison to other ZSL methods in Table 2 is unfair, since the baselines are trained on a smaller dataset and with different pretrained image and text encoders.
3. Although in Figures 1 and 3, the authors provide interesting evidence for the problem they identified, it is not clear how prevalent and important this problem is, especially considering the limited impact of the proposed method on the performance. Is it possible that only a handful of examples exist like those shown in the paper? Are there ways to quantify how common this problem is?
4. Although zero-shot recognition is a relatively new problem, learning cross-modality embeddings using image-caption datasets has been studied for a long time, with applications ranging from text-to-image retrieval to visual grounding. It is not clear whether the problem of image-caption alignment noise has been identified and studied before, or the authors are the first to identify this. It is also not clear how effective this method would be for such vision-language tasks besides zero-shot recognition.
5. In Table 1, it is strange that CLIP's performance using 400M images is close to the InfoNCE baselines trained on 3M images, on the GOI dataset. Is that only due to using pretrained image and text encoders? If so, it is important to elaborate the pretraining process, and compare the resources used for pretraining to CLIP's resources. It would also be interesting to show the performance without pretrained encoders, as a lower bound.


**Summary Of The Paper:**

It is common to train cross-modality embedding models on image-caption datasets using contrastive objectives, assuming each caption correctly describes the corresponding image, and all other captions can be used as negative (incorrect) matches. This paper clearly illustrates that this common assumption is wrong, and it hurts the performance of the learned embeddings on down-stream tasks, namely zero-shot recognition. To address this problem, the authors propose a simple and elegant approach to refine the pairwise image-caption matching, which can be seen as a smarter variation of label smoothing. This simple trick improves the performance of the well-known CLIP model consistently across datasets and backbone architectures.

**Summary Of The Review:**

The authors identify a new problem and propose a simple, elegant, and versatile solution to solve it, which has a consistent, incremental impact on the important task of zero-shot recognition. The insights and conclusions made by this paper are moderately significant, but not adequately supported, and not strong enough for acceptance.

UPDATE: after reading the authors' response, I have decided to raise my rating.

---

> ### Author Response · Authors · 2021-11-23
> **Response to reviewer qBzH**
>
> Thank you for your review and for appreciating our problem formulation and proposed solution.
>
> **Q: Are the improvements shown in table 1 significant?**
>
> First, we believe that our improvements in Table 1 are significant. Specifically, OTTER achieves on-average a 1.5% and a 0.6% improvement over the next closest baseline on GOI and IN10K, respectively.
>
> To better justify how significant a 2.3% performance gain is on the GOI and IN10K dataset (for RN50+Declutr-Sci-base). We provide a reference experiment where, instead of using OTTER to improve the performance, we resort to scaling up the model size and use a wide_ResNet50x2 (68.4M parameters / 11GFlops). For reference, ResNet50 has 25.6M parameters and 3.8GFlops.
>
> | Dataset | Method  | Image Encoder   | GOI F@K=1 | GOI F@K=5 | GOI F@K=10 | IN10K F@K=1 | IN10K F@K=5 | IN10K F@K=10 |
> |---------|---------|-----------------|-----------|-----------|------------|-------------|-------------|--------------|
> | CC 3M   | InfoNCE | wide_ResNet50x2 | 28.6      | 58.6      | 69.8       | 11.0        | 29.9        | 40.6         |
> |         | OTTER   | ResNet50        | 29.1      | 59.6      | 70.9       | 12.0        | 31.8        | 42.1         |
>
>
> Using an image encoder with more than twice the number of parameters, InfoNCE still underperforms OTTER on GOI and IN10K. This kind of performance gain is usually considered significant.
>
> **Q. What is the computational cost of OTTER?**
>
> First, OTTER is a training technique that does not increase any inference-time computational cost. In most of the efficient deep learning setting, people only care about inference time computational cost. In this sense, OTTER does not increase the compute cost of the trained model at all.
>
> For training, OTTER’s computational cost is also quite small. In our experiment, to train a RN50 + Declutr-Sci-base on CC3M for 10 epochs with 8 V-100 GPUs and a batch size of 512, it takes around 12 hours with InfoNCE and 16 hours with OTTER. Although OTTER requires 4 more hours than InfoNCE, 16 hours of compute cost is quite modest in computer vision research nowadays. In addition, the extra compute cost of OTTER mainly comes from the need to do forward pass twice (through both teacher and student models). This is needed by all the teacher-student distillation methods. The core-algorithm of OTTER, i.e., computing similarity targets using sinkhorn, etc., incurs negligible compute cost.
>
> **Q: How prevalent is the problem of many-to-many image and text matching in the datasets?**
>
> To illustrate the prevalence of this problem, we designed the following experiment:
> We use a CLIP VIT-B/32 pretrained on the 400M dataset to analyze how likely it is for an off-diagonal (non-default) image-text pair to match. We randomly sample 1000 batches from the CC3M and YFCC15M datasets, as we did in our training procedure, and use a pretrained CLIP to compute the image-to-text matching probabilities (by taking the dot-product of the feature embeddings and normalizing along each row). For each batch, we compute three statistics (averaged across rows):
> - On-diagonal (default)
> - Off-diagonal (non-default) average
> - Off-diagonal max
>
> The results below are the averages of 1000 batches.
>
> | Dataset  | Batch Size | On-diag | Off-diag Avg | Off-diag Max |
> |----------|------------|---------|--------------|--------------|
> | CC 3M    | 512        | 0.565   | 0.001        | 0.215        |
> |          | 1024       | 0.480   | 0.001        | 0.230        |
> |          | 2048       | 0.398   | 0.000        | 0.238        |
> | YFCC 15M | 512        | 0.628   | 0.001        | 0.197        |
> |          | 1024       | 0.551   | 0.000        | 0.219        |
> |          | 2048       | 0.469   | 0.000        | 0.239        |
>
> In the table above, we show that, on both CC and YFCC, there is a non-negligible probability that an off-diagonal pairing is also a good match, and the probability goes up if we use a larger batch size. The table above also shows that, with a batch size of 512, the prior on-diagonal matching probability is around 50-60%. This coincides with our ablation study, where $\alpha=0.5$ is the best choice among all other options.
>
> In addition, larger dataset size doesn’t positively correlate with higher noise level. In fact, larger size typically means more variety and lower chance of sampling similar or related captions/images in a single batch. In addition, YFCC 15M’s captions are in general longer and have more complicated structures than CC 3M. This explains the higher on-diagonal score estimated by CLIP.
>
> We also added more examples from both CC 3M and YFCC 15M to show that potential matching between non-paired images and captions are common. However, due to IP constraints, we can't directly use those images. We provided their URL and captions so reviewers can check out the examples themselves. This is the [link to the examples](https://docs.google.com/spreadsheets/d/1QhEUUh4w58G2I2oj9YD4TlVpPGWDdEw8_E3W3jSH-Vs/edit?usp=sharing)

---

> > ### Comment · Reviewer_qBzH · 2021-11-29
> > **Feedback**
> >
> > The authors have addressed most of my concerns, and I appreciate their thorough response to my questions. I have decided to raise my rating to positive.

---

> ### Author Response · Authors · 2021-11-23
> **Response to reviewer qBzH (part 2)**
>
> **Q: Comparison with other ZSL methods in table 2 is unfair.**
>
> The comparison with the traditional ZSL methods in table 2 is to provide readers a reference and put OTTER in the context of traditional ZSL methods (with lower performance than us) and modern ones such as CLIP (with stronger performance than us). We do not claim that our methods achieve better results only because of OTTER. In fact, we use more data and contrastive learning, etc.; all these contribute to better results. Our main claim is that OTTER is better than InfoNCE, which is supported by the results in Table 1.  In our paper revision, we have removed the claims of beating traditional ZSL methods in the abstract and introduction to further eliminate the confusion that we directly compare with previous ZSL methods in a similar setting.
>
>
> **Q: Has image-caption alignment noise been identified and studied before? How effective is OTTER for vision-language tasks besides zero-shot recognition?**
>
>  As far as we know, we are the first to observe and study the noisy image-text matching issue in V+L contrastive learning. Please see the response to Reviewer Qrbe (“Can you emphasize the difference between existing optimal transport based methods?”) for a comparison with existing works we are aware of.
>
> Our paper is solely focused on zero-shot image classification and other V+L tasks are out of the scope of our paper.
>
> **Q: Effect of pretraining**
>
> First of all, InfoNCE, Knowledge Distillation, and Label Smoothing are baselines of OTTER. We don’t mean to compare with CLIP’s results using 400M image-text pairs, since the datasets, training procedures, model architectures etc. are all different.
>
> Our experimental settings are clearly reported in the paper, and we directly used open-source pretrained models from timm [https://github.com/rwightman/pytorch-image-models] and hugging face [https://huggingface.co] in our experiments. Without using pretrained weights, the model converges much more slowly, and using 10 epochs of training, models don’t fully converge and could only achieve ~12% GOI F@K=1. We could not conduct any meaningful comparison based on this. Achieving a reasonable performance (similar to the numbers reported in our paper) without pretraining would require significantly more epochs. So as a design decision, we conduct all our experiments using pretrained weights to save compute resources. For results of OTTER on YFCC15M, please see response to Reviewer DJW4 (“Does OTTER work on larger-scale datasets?”).

---

### Official Review · Reviewer_Qrbe · 2021-11-03

**Correctness:** 3
**Technical Novelty And Significance:** 3
**Empirical Novelty And Significance:** 3
**Recommendation:** 6
**Confidence:** 4

**Main Review:**

Pros:
+ The motivation of this paper is very clear.
+ Overall, the paper is well written. In particular, the INTRODUCTION section has a nice flow.

Cons:

 +For the optimal transmission algorithm, in the METHODS part, the author should emphasize the difference from the existing algorithm.

 +More visualizations should be added to analyze the effectiveness of the proposed method (E.g Figure 2).

 +For IN10K FH@K, the proposed OTTER does not achieve satisfactory performance. The author should analyze the reason for this result.


 Minor comments:

- In the 4th line in 4.2 ABLATION STUDIES, it should be "define" instead of " defined ".

**Summary Of The Paper:**

Review: This paper proposes a novel framework, OTTER, which considers the many-to-many relationship within a batch of images and text captions for data-efficient language-supervised zero-shot recognition. An improved InfoNCE is explored to consider the many-to-many relationship between unpaired images and texts. Then to further explore the relationship between images and texts, the optimal transmission method is introduced. Extensive experiments on six popular benchmark datasets demonstrate that OTTER well addresses this challenge, and it only needs fewer samples.


**Summary Of The Review:**

Due to the limited novelty of the proposed method, I need to re-evaluate my judgments after the rebuttal.

UPDATES: The authors have addressed all of my concerns, though experiments on 400M dataset are missed by uncontrollable reasons, as initial judgement, I tend to accept this paper.

---

> ### Author Response · Authors · 2021-11-23
> **Response to reviewer Qrbe**
>
> Thank you for your review and for acknowledging that "the motivation is very clear" and "the paper is well written".
>
> **Q: Can you emphasize the difference between existing optimal transport based methods?**
>
> To the best of our knowledge, we are aware of two papers [1, 2] in vision and language that adopts optimal transport in the contrastive learning setting. Fundamentally, however, OTTER is solving a different problem than these works.
> * [1, 2] aim to ground words in a sentence to regions in an image. Given a paired image and text caption, there are no labels telling which words correspond to which regions in the image, [1, 2] use optimal transport to find such word-to-region matches.
> * In comparison, OTTER addresses the noisy matching issue between images and texts in a batch for contrastive learning. We use optimal transport to re-match an image to other suitable text captions, and a text to suitable images.
> * [3] uses an additional OT-based Wasserstein loss in contrastive representation distillation. The loss matches student representations to teacher representations in a batch. [3] is different from our method since it directly minimizes the Wassertein loss between two models’ representations, while our method uses OT to estimate the pairing probability and use the probability for knowledge distillation.
>
> If reviewer Qrbe has other specific papers in mind, we would appreciate it if reviewer Qrbe could point those papers to us and we can further clarify our differences.
>
> Ref:
>
> [1] Graph optimal transport for cross-domain alignment
>
> [2] UNITER
>
> [3] Wasserstein Contrastive Representation Distillation
>
> **Q: Why does the performance on IN10K seem unsatisfactory?**
>
> First, on IN10K, OTTER still outperforms all other baselines in 13 out of 18 settings, with an average improvement of 0.6% over the next closest baseline, including cases when OTTER underperforms. As more evidence, in response to Reviewer DJW4(“Does OTTER work on larger scale datasets”), we show that OTTER is also able to outperform InfoNCE on both GOI and IN10K when trained with YFCC 15M. In general, OTTER still clearly outperforms compared the three baselines.
>
> Please also see response to Reviewer qBzH (“Are improvements in table 1 significant?”). If we aim to achieve such improvements through scaling up the model instead of using OTTER, we would have to use a wide_ResNet50x2 with  2.8x as many FLOPs and 2.6x as many parameters. Considering this, we argue that improvement achieved by OTTER is nontrivial.
>
> **Q: Are there more visualizations to analyze the effectiveness of OTTER?**
>
> We would appreciate if Reviewer Qrbe could give more specific suggestions on what kind of visualizations Reviewer Qrbe has in mind. We also added more examples from both CC 3M and YFCC 15M to show that potential matching between non-paired images and captions are very common. However, due to IP constraints, we cannot directly use those images for visualization. We provided their URL and captions so reviewers can check those examples out themselves. This is the google doc [link to the examples](https://docs.google.com/spreadsheets/d/1QhEUUh4w58G2I2oj9YD4TlVpPGWDdEw8_E3W3jSH-Vs/edit?usp=sharing).

---

> > ### Comment · Reviewer_Qrbe · 2021-11-30
> > **Additional issues**
> >
> > In my opinion, the proposed OTTER scheme should be trained on the 400M dataset (CLIP used), only in this way, we can investigate directly the performance improvements compared with CLIP.
> >
> > However, the authors said CLIP has not released such 400M dataset. By contrast, OTTER do all their experiments by using 3M (CC) dataset together with some kinds of pre-trained image/text encoders.  Here, my questions are
> >
> > 1) are the image encoders (e.g, ResNet50, ResNet34, and etc. listed in Table 1 and 2) pretrained on 400M or just imagenet dataset?
> >
> > 2) in table 1 and 2, under the training data of 3M (CC) and 5M （WIT）, are the methods of InfoNCE actually CLIP method -- i.e., train the models on such small datasets by Eq. (1)?

---

> > > ### Author Response · Authors · 2021-11-30
> > > **Follow-up to Reviewer Qrbe**
> > >
> > > Thanks again for the review. First, I would like to point out that CLIP trained on 400M data is not our baseline, but the underlying algorithm of CLIP, InfoNCE is our baseline. The experiments in table 1 provide a direct comparison of OTTER and InfoNCE under the same setting(identical training recipe and initialization etc.) Indeed, it's not possible to directly train on the 400M dataset since it's not released, we have added new results to show that OTTER also outperforms InfoNCE on YFCC 15M (subset of YFCC100M). Please see Reviewer DJW4(“Does OTTER work on larger scale datasets”).
> > >
> > > Regarding your questions:
> > > 1. The image encoders for OTTER and the baselines in Table 1 and 2 are all initialized with weights pretrained on ImageNet(1.4M images). We directly used pretrained weights from the timm github repo.
> > > 2. Yes, InfoNCE is the algorithm used by CLIP. The only differences with our InfoNCE baseline and CLIP are that
> > >  - we initialize with pretrained image and text encoders
> > >  - we use different training recipes (smaller batch size and follow the training recipe of BiT)

---

### Decision · Program_Chairs · 2022-01-20

**Decision:**

Accept (Poster)

**Comment:**

The paper addresses the interesting  many-to-many assignement problem between a set of images and a set of text. Most reviewers, (and I agree with them) think that the  idea and its application worth being published although the performance improvement
is marginal. I request the authors to update the paper based on the discussions.